



# On the accuracy of the measured and modelled surface latent and sensible heat flux in the interior of the Greenland Ice Sheet

Ida Haven[1,2], Hans Christian Steen-Larsen[1,3], Laura J. Dietrich[1,3], Sonja Wahl[1,3,4], Jason E. Box[5], Michiel R. van den Broeke[2], Alun Hubbard[6,7], Stephan T. Kral[1,3], Joachim Reuder[1,3], and Maurice van Tiggelen[2]

[1]Geophysical Institute, University of Bergen, Bergen, Norway
[2]Institute for Marine and Atmospheric research Utrecht, Utrecht University, Utrecht, The Netherlands
[3]Bjerknes Centre for Climate Research, Bergen, Norway
[4]Laboratoire des Sciences du Climat et de l'Environnement, IPSL-CEA-CNRS-UVSQ, Université Paris-Saclay, Gif-sur-Yvette, France
[5]Geological Survey of Denmark and Greenland (GEUS), Copenhagen, Denmark
[6]Geography Research Unit, University of Oulu, Oulu, Finland
[7]Centre for Ice, Cryosphere, Carbon & Climate, Institutt for Geovitenskap, UiT - the Arctic University of Norway, Tromso

**Correspondence:** Ida Haven (idahaven@kpnmail.nl) and Hans Christian Steen-Larsen
(Hans.Christian.Steen-Larsen@uib.no)

**Abstract.** The latent (LHF) and sensible (SHF) heat fluxes are key components of the surface mass and energy balance in the accumulation area of the Greenland Ice Sheet, making them critical for accurate sea level projections. While Eddy-Covariance (EC) systems provide accurate measurements of the turbulent surface transport of mass and energy in the low and mid-latitudes, frequent stable boundary layer conditions in polar regions introduce uncertainties in the EC method. In addition, as EC mea-

surements are sparse, it is critical to characterise biases in the more common bulk fluxes obtained from automatic weather stations and climate models in polar areas. In this study, we present an intercomparison of three independent EC systems at the EastGRIP site at $\sim 2700\,\mathrm{m}$ a.s.l on the Greenland Ice Sheet to assess the accuracy of LHF and SHF measurements. A comparison of the fluxes by the three systems demonstrates excellent agreement, with a correlation ($r$) of 0.97 to 0.98, an absolute bias of $0.2\,\mathrm{W\,m^{-2}}$, an $RMSE$ between $1.2\,\mathrm{W\,m^{-2}}$ and $1.5\,\mathrm{W\,m^{-2}}$ and slopes between 1.01 and 1.16 for the LHF,

and $r = 0.98$, an absolute bias of less than $0.5\,\mathrm{W\,m^{-2}}$, an $RMSE$ between 1.6 and $1.9\,\mathrm{W\,m^{-2}}$, and slopes of 1.0 for the SHF. A comparison of the validated EC fluxes against the bulk method highlights the sensitivity to the site-specific roughness length $z_{0,m}$ and the limitation of common parameterisations of the humidity and temperature roughness lengths $z_{0,q}$ and $z_{0,t}$. Using improved values for $z_{0,m}$, $z_{0,q}$ and $z_{0,t}$, recomputed bulk fluxes are compared to fluxes simulated by regional climate models MAR, RACMO2.3p2 and RACMO2.4p1. We find an overall good agreement of the summer turbulent flux magnitudes,

while all evaluated models simulate stronger near-surface temperature gradients during winter compared to observations from automatic weather stations, leading to consistently larger modelled SHF and LHF values in winter.



## 1    Introduction

The Greenland Ice Sheet is currently the main single contributor to global ocean mass changes (Otosaka et al., 2023) and, together with the Antarctic Ice Sheet, the biggest source of uncertainty in predicting future sea level rise (Fox-Kemper et al., 2021). Aschwanden et al. (2019) and Plach et al. (2019) show that one of the main uncertainties in accurately simulating past variations and future projections of the volume of the ice sheet is related to inaccuracies of the surface mass balance (SMB). The turbulent latent heat flux (LHF) directly impacts the SMB by adding or removing water through evaporation/sublimation and condensation/deposition and linking the mass balance to the surface energy balance (SEB). A thorough understanding of the ice sheet SEB and its impact on SMB is therefore fundamental, particularly across the interior regions of the Greenland Ice Sheet where annual precipitation rates are low and the LHF can act seasonally as a key driver in the SMB (Dietrich et al., 2024). Furthermore, recent research on ice core climate reconstruction (e.g. Dietrich et al., 2023; Wahl et al., 2022) has documented the importance of having access to accurate observations and simulations of LHF for quantifying the impact of post-depositional processes on the water isotope climate signal in the snow. This illustrates the broad need for reliable observations and simulations of the SMB across the Greenland and Antarctic Ice Sheet.

Regional climate models (RCMs) can provide year-round simulations of the surface turbulent heat fluxes. However, it has been documented that they struggle to represent the sensible (SHF) and latent heat flux components of the surface mass balance appropriately (Noël et al., 2018; Dietrich et al., 2024). As a result of quasi-permanent radiative cooling of the interior ice sheet surface, the atmospheric surface layer is usually stably stratified and shallow, and characterized by low temperatures and humidity content and a relatively smooth snow surface. The resulting low magnitude of the turbulent fluxes and the occurrence of steep gradients of temperature, moisture and wind in the atmospheric surface layer, make it an extremely challenging environment for the RCMs. Model evaluations of turbulent heat fluxes in the accumulation zone of the ice sheet are scarce (Ettema et al., 2010), mainly due to the lack of robust and accurate direct observations of the turbulent exchange in the atmospheric surface layer (Miller et al., 2018), resulting in a standstill in the models' skill to represent the turbulent fluxes. Besides their spatial and temporal scarcity, it is the high uncertainty of turbulent flux measurements over the ice sheet themselves that poses a major challenge to improving the models' representations.

The established standard method for measuring turbulent fluxes is based on the eddy covariance (EC) technique (Swinbank, 1951; Mauder et al., 2021). Wind and temperature fluctuations are typically measured at high temporal resolution ($10 - 50\,\mathrm{Hz}$) using a sonic anemometer, along with humidity variations using an open or closed path water vapour analyser. The SHF and the LHF can be directly calculated as the covariance of vertical velocity fluctuations, and the corresponding temperature and humidity fluctuations, respectively. Yet, the considerable logistic requirements with respect to reliable (over-winter) power supply, supervision and maintenance, allow only rather limited deployments in remote areas under such extreme, polar meteorological conditions. Corresponding EC data sets for the ice sheet are therefore sparse (Van Tiggelen et al., 2021, 2024) and generally limited to summertime deployments.

Year-round observational estimates of SHF and LHF can be obtained by using bulk methodology (e.g. Sun et al., 1999), based on flux-gradient relationships formulated by application of Monin-Obukhov similarity theory (MOST). Mean values of wind





speed, temperature and humidity measured by an automatic weather station (AWS), combined with additional information or assumptions on surface temperature and surface humidity, and empirical, stability dependent drag coefficients, based on MOST, allow for an estimation of SHF and LHF even if meteorological variables are only available from one level. The existing limitations in the validity of MOST, in particular for moderately and strongly stable atmospheric conditions (e.g.,

Grachev et al., 2005; Cullen et al., 2007; Schlögl et al., 2017; Pfister et al., 2019), which are common on top of the Greenland Ice Sheet during winter (Cullen and Steffen, 2001), are expected to increase uncertainty for corresponding SHF and LHF bulk estimates. This is also corroborated by studies that found that the bulk method leads to an underestimated LHF both at the edge of the Greenland Ice Sheet (Box and Steffen, 2001) and on the Antarctic Ice Sheet (Town and Walden, 2009). Additional sources of uncertainty are the dependency of the bulk method on the surface roughness length, which is highly variable across

the ice sheet (Van Tiggelen et al., 2021), and indications of flux underestimation by the bulk method in conditions of drifting and blowing snow (Sigmund et al., 2022).

Despite these problems, the bulk method is to date fundamental for the year-round estimation of turbulent heat fluxes, both by meteorological measurements and model simulations. The LHF can in theory also be directly measured by an evaporation pan (Box and Steffen, 2001), however, the method is time-consuming to execute and prone to issues like excessive heating,

limited snowfall and blowing snow. A simultaneous direct determination of LHF and SHF can only be provided by the EC technique. Although tested and proven under non-polar conditions (Mauder and Zeeman, 2018; Wang et al., 2016; Loescher et al., 2005; Schmidt et al., 2012; Polonik et al., 2019), an intercomparison of multiple independent EC systems at a polar site, in particular focusing on the LHF accuracy, has not been conducted to date.

The aim of this study is to address this knowledge gap, and further, to determine the accuracy and validate EC measurements

at a high elevation on a polar ice sheet. We achieve this by conducting a comparison of the near-surface LHF and SHF measured by three co-located EC systems during the summer of 2019 at the EastGRIP fieldsite and comparing our results with other EC intercomparison studies. The validated EC measurements are then used to provide an observationally-based value for the roughness length at the site. This yields an improved estimate of the turbulent heat fluxes based on the bulk method. This improved year-round record of bulk estimates from 2016 to 2020 is then compared to the output from the high-resolution

Regional Atmospheric Climate Model (RACMO) and Modèle Atmosphérique Régional (MAR).

## 2   Data

### 2.1   Measurement site

All data used in this study has been collected at the location of the East Greenland Ice Core Project (EastGRIP) field site ($75°37'47''$ N, $35°59'22''$ W). EastGRIP is located in the north-eastern part of the Greenland Ice Sheet (Fig. 1a) approximately

$350\,\mathrm{km}$ NNE of Summit, the highest point of the ice sheet. The local time is set to UTC-2 and UTC-1 during daylight saving time. In this study, all times are given in UTC and the sign convention is positive for upward fluxes, corresponding to sublimation for the LHF. The observational data comes from three EC systems, installed upwind of the EastGRIP camp, an AWS that is part of the Programme for Monitoring of the Greenland Ice Sheet (PROMICE, station name "EGP", Fausto et al.,





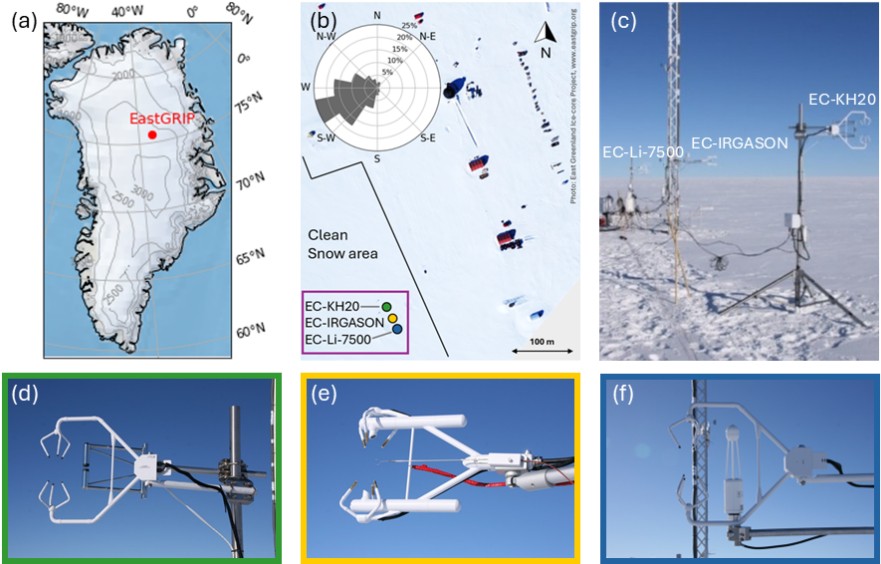

**Figure 1.** (a) Location of the EastGRIP fieldsite on the Greenland Ice Sheet ($75°37'47''$ N, $35°59'22''$ W), (b) camp overview with the location of the EC-systems in the dedicated clean snow area upwind of the camp along with the windrose during the campaign, (c) overview picture of the three individual EC systems oriented perpendicular to the prevailing wind direction, and detailed views of the individual EC setups (d) CSAT3+KH20 (EC-KH20), (e) IRGASON (EC-IRGASON), (f) CSAT3+Li-7500 (EC-Li-7500). Surface elevation in Fig. 1a is from Morlighem et al. (2017).

2021), and an AWS that is part of GC-Net (station name "EastGRIP", Vandecrux et al., 2023). Both AWSs are located in the
vicinity of the EastGRIP camp with sufficient distance to rule out flow disturbance by the campsite (approximately $500\,\mathrm{m}$ and $1000\,\mathrm{m}$ distance for the GC-Net and PROMICE AWS, respectively). The three EC systems were installed at $2.15\,\mathrm{m}$ height in a dedicated clean-snow area, set approximately $6\,\mathrm{m}$ apart perpendicular to the dominant wind direction, with the instruments facing into the wind (Fig. 1b and 1c).

We present observational data from the three EC systems sampled during the period from the 28th of May 2019 until the
31st of July 2019. During this period, the PROMICE AWS recorded a mean air temperature of $-10.2\,°\mathrm{C}$, ranging from $-25.9$ to $1.3\,°\mathrm{C}$ and a mean air pressure of $730\,\mathrm{hPa}$. The mean windspeed was $4.5\,\mathrm{m\,s^{-1}}$, with a maximum of $13.7\,\mathrm{m\,s^{-1}}$ and an average direction of $254\,°$ (Fig. 1b). The specific humidity was on average $2.2\,\mathrm{g\,kg^{-1}}$, ranging from $0.5\,\mathrm{g\,kg^{-1}}$ to $5.0\,\mathrm{g\,kg^{-1}}$. A model comparison is done for the 4-year period from 2016-2019. During this period the PROMICE AWS recorded an average temperature of $-27.6\,°\mathrm{C}$, ranging from $-63.3\,°\mathrm{C}$ to $1.3\,°\mathrm{C}$, a mean air pressure of $720\,\mathrm{hPa}$ and an average windspeed of
$5.3\,\mathrm{m\,s^{-1}}$, with a maximum of $21.3\,\mathrm{m\,s^{-1}}$. The average wind direction during this 4-year period was $242\,°$.





## 2.2 Eddy-Covariance systems

The data for the EC comparison comes from three EC setups. The first EC system is a combination of a CSAT3 sonic anemometer (Campbell Scientific) and a Krypton Hygrometer 20 (KH20, Fig. 1d, Campbell Scientific). The second EC system is the IRGASON (Campbell Scientific), which is a combined sonic anemometer and open-path gas analyser (Fig. 1e). The third system is the combination of a CSAT3 sonic anemometer and a Li-7500 (LI-COR) open-path gas analyser (Fig. 1f). To measure the humidity fluctuations, the IRGASON and Li-7500 both use infrared absorption, while the KH20 uses a krypton laser. The IRGASON uses radiation at $2.7\,\mu m$ (absorption) and $2.3\,\mu m$ (reference); the Li-7500 uses absorption at $2.59\,\mu m$ and $3.95\,\mu m$ as a reference; the KH20 uses $123.58\,nm$ along with a minor light at $116.49\,nm$, both are absorbed by $H_2O$ and $O_2$, and $H_2O$ concentration is computed from the combined signal of the two wavelengths. The combined systems of sonic anemometer and hygrometer will be referred to as EC-KH20, EC-IRGASON, and EC-Li-7500. The EC-IRGASON and EC-Li-7500 ran from the 28th of May 2019 until the 31st of July 2019. The EC-KH20 was deployed for three fewer days and ran from the 29th of May until the 29th of July 2019. During this period the EC systems sampled continuously at a frequency of 20 Hz. During the days around the 29th of June and the 1st of July, a combination of snowfall and wind led to measurement issues and consequent gaps in the time series. On the night of the 17th to the 18th of July, possible frosting of the instruments led to a gap in the time series from 2019-07-18 00:00 until 2019-07-18 09:00. The KH20 has a gap in the data from 2019-06-02 14:00 until 2019-06-03 20:00. The EC-KH20 was also deployed in the summers of 2016, 2017 and 2018 (Steen-Larsen et al., a, b, 2022). Due to the longer data availability, the EC-KH20 flux observations from the summers of 2016 to 2019 are used to compare with the model simulations.

## 2.3 Automatic weather stations

In this study data from two AWSs is used: the PROMICE AWS (Fausto et al., 2021; How et al., 2022) and the GC-Net AWS (Vandecrux et al., 2023; Steffen et al., 2022). The data from the PROMICE AWS is used to compute the LHF and SHF with the one-level bulk method. The GC-Net data is only used for comparison with the PROMICE data to assess the data quality during winter.

The PROMICE AWS was installed on the 1st of May 2016 and has been collecting data uninterruptedly ever since. It measures single-level temperature, wind speed, relative humidity and up- and downwelling shortwave and longwave radiation at approximately $2\,m$ height above the surface, varying between 1.8 and $2.6\,m$ due to changes in snow height. Instrument specifications and uncertainties are described in Fausto et al. (2021). For the model comparison, it is assumed that the height is close enough to the $2\,m$ model output, that it does not need to be corrected (similar to in Dietrich et al., 2024). For specific case studies, the exact measurement height will be provided. The surface temperature is determined using the up- and downwelling longwave radiation and an emissivity of 0.97 (Fausto et al., 2021). The surface specific humidity is determined using the surface temperature and assuming saturated conditions. Inspection of the relative humidity, temperature and longwave radiation data obtained from the PROMICE station during the winter indicates unrealistic values at temperatures below $-50\,°C$, i.e. that the relative humidity fully depends on temperature and that occurrences of low temperature (below $-50\,°C$) take place without





simultaneous radiative cooling. Thus, the data with temperatures below $-50\,^{\circ}\mathrm{C}$ is removed. Further discussion of the data
quality in winter is provided in Sect 5.3.

The GC-Net AWS measures air temperature, relative humidity and wind speed at two vertical levels typically separated by
$1.2\,\mathrm{m}$ (instrument specification and uncertainties described in Vandecrux et al., 2023). The distance of the lowest level to the
snow surface varied between $0.4\,\mathrm{m}$ and $2.3\,\mathrm{m}$ during the years 2016 to 2019 owing to snow accumulation, compaction and
ablation. Instrument heights are provided in the relevant figures.

## 2.4   RACMO

This study uses two different polar versions of RACMO, a hydrostatic model that combines the atmospheric dynamics of
the High Resolution Limited Area Model (HIRLAM, version 5.0.3) with the physics package from the European Centre for
Medium-range Weather Forecasts (ECMWF) Integrated Forecast System (IFS) (Noël et al., 2018; Van Dalum et al., 2024).
The polar version of RACMO is developed and maintained at the Institute for Marine and Atmospheric Research Utrecht
(IMAU) and uses specialised parameterisations to simulate the Arctic and Antarctic climate. Over the Greenland Ice Sheet and
surroundings, RACMO is run on a $5.5\,\mathrm{km}$ resolution, with 40 vertical layers. The simulations from both model versions are
forced with 3-hourly ERA-5 reanalysis at the lateral boundaries and for both simulations, the output from the gridpoint closest
to EastGRIP is analysed.

The oldest version of RACMO used in this study is RACMO 2.3p2 (Noël et al., 2018, hereafter referred to as RACMO2.3),
from which the three-hourly model output is analysed. In RACMO2.3 the ECMWF-IFS physics cycle CY33R.1 is used (Noël
et al., 2018). The LHF and SHF are simulated using the bulk method, a constant roughness length for momentum ($z_{0,m}$) of
$1 \times 10^{-3}\,\mathrm{m}$ and the Andreas (1987) parameterisation to determine the roughness lengths for moisture ($z_{0,q}$) and heat ($z_{0,t}$)
over snow surfaces. In RACMO2.3, the sublimation from blowing snow is included in the surface LHF.

The second version of RACMO is RACMO2.4p1 (Van Dalum et al., 2024, hereafter referred to as RACMO2.4), from which
the hourly model output is analysed. One major difference with RACMO2.3 is that in RACMO2.4 the ECMWF-IFS physics
package is updated to physics cycle CY47R.1 (Van Dalum et al., 2024) and the blowing snow scheme is improved, as described
in Gadde and Van de Berg (2024). The LHF and SHF in RACMO2.4 are simulated in the same way as RACMO2.3. However,
sublimation from blowing snow is no longer directly included in the LHF but is stored as a separate variable and only the
surface sublimation is evaluated in this study. The implications of how blowing snow sublimation is handled in the model
simulations are discussed in Sect. 4.3.

## 2.5   MAR

In a similar study by Dietrich et al. (2024), turbulent flux simulations with the RCM MAR (e.g., Fettweis et al., 2017) were
evaluated for the EastGRIP site using improved turbulent flux estimates from the PROMICE weather station and the EC-KH20.
We show these results alongside the results for the RACMO evaluation in this study. The model data comes from the gridpoint
closest to EastGRIP.



In all figures, MAR data (Dietrich, 2023) is presented for the same periods as for RACMO, from between 2016 and 2019. MAR was run on a $30\,\mathrm{km}$ horizontal and 30 vertical layer resolution, which is substantially coarser than in the RACMO simulation. Nonetheless, due to the smooth orography of the accumulation area of the Greenland Ice Sheet, we consider a direct comparison between both simulations for MAR and RACMO valid. As in RACMO, the MAR simulation is forced by the ERA-5 reanalysis product and turbulent fluxes are calculated using a one-layer bulk method with roughness lengths for moisture and heat calculated from the roughness length of momentum following Andreas (1987). However, it should be noted that a constant roughness length of momentum of $1.3 \times 10^{-4}\,\mathrm{m}$ was used for the MAR simulation, much lower than the value of $1 \times 10^{-3}\,\mathrm{m}$ used in the RACMO simulation. Consequently, we would expect lower turbulent flux amplitudes for the MAR output compared to RACMO.

## 3 Methods

### 3.1 Eddy-Covariance flux computation

The calculation of fluxes using the EC method requires compliance with EC flux theory and adequate instrumentation. The EC flux technique assumes that transport is turbulence-dominated, with negligible contributions from (sub)meso motion timescales. The method further assumes homogeneity of the terrain, absence of convergence or divergence and stationarity. As the terrain upwind of the EC systems is flat and homogeneous, the EastGRIP field site is well suited for EC measurements. Thus, topography-driven transport regimes are assumed to be negligible and the transport therefore dominated by turbulence. The EC instrumentation was set up at $2\,\mathrm{m}$ height to ensure measurements were performed in the atmospheric surface layer, which can be shallow in polar, snow-covered conditions.

The EC data is processed using the TK3 software from the University of Bayreuth (Mauder and Foken, 2015). The raw data is filtered using consistency limits of $-50$ to $50\,\mathrm{m\,s^{-1}}$ for horizontal wind speed ($u$), $-10$ to $10\,\mathrm{m\,s^{-1}}$ for vertical wind velocity ($w$) and $-80$ to $30\,^{\circ}\mathrm{C}$ for air temperature ($T$) and spikes are removed using the MAD spike test (Mauder et al., 2013) applying a standard deviation of 3.5. For the EC-KH20 a high number of simultaneous spikes was detected in the measured T and w. These spikes have been removed based on the quality flags provided by the CSAT3. As the focus is on getting accurate fluxes rather than having a complete record, no interpolation is done for the missing values. The raw covariances are calculated using a 10-minute averaging time, cross-correlation and planar fit correction (Wilczak et al., 2001) and are corrected using the Moore (Moore, 1986), WPL (Webb et al., 1980) and the Tanner correction (Tanner et al., 1993) for the EC-KH20. The 10-minute averaging time is chosen based on a modified version (using $30\,\mathrm{min}$ instead of $4\,\mathrm{h}$) of the convergence test by Foken (2006) to ensure the averaging time is suitable for the conditions, i.e. that it is long enough to capture all turbulent variations but does not include (sub)mesoscale effects. The Moore (1986) correction is applied to correct for spectral losses and uses a transfer function based on the Kaimal spectrum (Kaimal et al., 1972), derived from the 1968 Kansas experiment. The use of the Kaimal spectrum for the correction of spectral losses is validated by comparing the shape of $w$, $T$, $a$, $w'T'$ and $w'a'$ (co)spectra to the shape of the Kaimal spectrum (S1 in the supplementary material). After the processing with the TK3 software, remaining outliers in the average flux data are removed by excluding time intervals where $u$ was smaller than $1\,\mathrm{m\,s^{-1}}$ or larger than





$8\,\mathrm{m\,s^{-1}}$, or when the absolute value of $w$ was larger than $0.15\,\mathrm{m\,s^{-1}}$ or the wind direction was between $20\,^\circ$ and $110\,^\circ$ (the direction in which the mounting installation obstructed the flow). Final outliers in the average flux data are removed based on visual inspection. This leaves $84\,\%$ of the 10-min LHF and SHF intervals for the EC-IRGASON, $85\,\%$ of the intervals for the EC-Licor-7500 and $86\,\%$ of the intervals for the EC-KH20. An overview of the time series and the data removed in post-processing is provided in the Supplementary Material (Fig. S2).

### 3.2   Bulk sensible and latent heat flux computation

To obtain year-round measurements of the LHF and SHF to compare to the regional model output, the turbulent heat fluxes are calculated from the PROMICE AWS observations using the bulk method. We evaluate both the bulk-calculated LHF and SHF provided in the published PROMICE data product (Fausto et al., 2021), as well as re-calculated LHF and SHF using site-specific roughness lengths, against the EC data in summer. Similar to Fausto et al. (2021), we calculate the SHF and LHF using the one-level bulk method, with equations 4 and 5 described in Van As (2011) (note the added minus results from the sign convention used in this study):

$$\mathrm{SHF} = -\rho C_p \kappa^2 \frac{u}{\ln\frac{z_u}{z_{0,m}} - \psi_u} \frac{T - T_\mathrm{s}}{\ln\frac{z_t}{z_{0,t}} - \psi_t} \tag{1a}$$

$$\mathrm{LHF} = -\rho L_\mathrm{s} \kappa^2 \frac{u}{\ln\frac{z_u}{z_{0,m}} - \psi_u} \frac{q - q_\mathrm{s}}{\ln\frac{z_q}{z_{0,q}} - \psi_q} \tag{1b}$$

Here $\rho$ is the air density, $Cp$ the specific heat capacity of air at constant pressure, $k = 0.4$ the Von Kármán constant, $u$ the wind speed, $T$ and $T_s$ the temperature measured at $2\,\mathrm{m}$ and at the surface, respectively, $L_s$ the latent heat of sublimation, and $q$ and $q_s$ are the corresponding specific humidities. The surface specific humidity is derived from the surface temperature following the equations in Murphy and Koop (2005), using the equilibrium vapour pressure over ice and assuming saturated conditions. $z_u$, $z_t$ and $z_q$ are the measurement heights of the wind speed, temperature and humidity and $z_{0,m}$, $z_{0,t}$ and $z_{0,q}$ the roughness lengths of momentum, heat and moisture. $\psi_u$ and $\psi_q$ are the stability correction functions for momentum and moisture, following Holtslag and De Bruin (1988) under stable conditions and the Businger–Dyer expressions, described in Paulson (1970), under unstable conditions. $\psi_t$ and $\psi_q$ are assumed to be the same.

Stable conditions:

$$\psi_u = -(0.7\zeta + 0.75(\zeta - \frac{5}{0.35})\exp(-0.35\zeta) + 0.75\frac{5}{0.35}) \tag{2a}$$

$$\psi_q = \psi_u \tag{2b}$$





Unstable conditions:

$$\psi_u = 2\ln\left(\frac{1+x}{2}\right) + \ln\left(\frac{1+x^2}{2}\right) - 2\tan^{-1}(x) + \frac{\pi}{2} \tag{3a}$$


$$\psi_q = 2\ln\left(\frac{1+x^2}{2}\right) \tag{3b}$$

Where $\zeta = z/L$, $L$ is the Monin-Obukhov length, $x = (1 - \frac{\gamma}{\zeta})^{1/4}$ and $\gamma = 16$ is an empirically derived constant (Paulson, 1970).

The bulk flux formulations require a value for the roughness lengths. For the bulk calculation done by PROMICE a fixed
$z_{0,m} = 1 \times 10^{-3}$ m is used for the entire Greenland Ice Sheet together with the parameterisation of Smeets and Van den Broeke (2008a, b) (S&vdB) to determine $z_{0,q}$ and $z_{0,t}$, which are assumed equal. Literature values of roughness lengths above a snow surface cover a wide range of $1 \times 10^{-1}$ m - $1 \times 10^{-5}$ m (Van Tiggelen et al., 2021; Smeets and Van den Broeke, 2008b). To ensure a robust comparison between bulk and EC methods, we use two different methods for determining $z_{0,m}$, $z_{0,t}$ and $z_{0,q}$. The first uses $z_{0,m}$ based on our EC measurements and the Andreas (1987) parameterisation for $z_{0,q}$ and $z_{0,t}$, which are assumed
equal. The second method uses $z_{0,m}$, $z_{0,t}$ and $z_{0,q}$ obtained from the EC measurements. The data from the EC-IRGASON is used, because this system requires no separation correction and provides the most accurate humidity measurements (see Sect. 4.1 and S1 in the supplementary material). It is assumed that the turbulence properties are similar for the other setups, due to the limited separation of 6 m and the homogeneous terrain. The values for $z_{0,m}$, $z_{0,t}$ and $z_{0,q}$ are estimated using Monin-Obukov similarity and calculated from the measured EC-IRGASON fluxes and the vertical gradients from the PROMICE AWS
following Van Tiggelen et al. (2023):

$$z_{0,m} \simeq \frac{z}{\exp\left(\kappa \frac{u(z)}{u_*} + \psi_m\left(\frac{z}{L}\right)\right)}, u_* = \left(\overline{u'w'}^2 + \overline{v'w'}^2\right)^{1/4} \tag{4a}$$

$$z_{0,q} \simeq \frac{z}{\exp\left(\kappa \frac{a(z)-a_s}{a_*} + \psi_m\left(\frac{z}{L}\right)\right)}, a_* = -\overline{w'a'}/u_* \tag{4b}$$

$$z_{0,t} \simeq \frac{z}{\exp\left(\kappa \frac{T(z)-T_s}{T_*} + \psi_h\left(\frac{z}{L}\right)\right)}, T_* = -\overline{w'T'}/u_* \tag{4c}$$

Here $z$ is the height of the EC or PROMICE instruments above the snow surface, which are both 2.15 m during the measurement campaign in 2019. $u(z)$ is the 2 m PROMICE wind speed and $u_*$ is the friction velocity, the value of which is based on the EC-IRGASON measurements and provided by TK3. The stability functions are the same as for the bulk calculation. $w'a'$ is the EC-IRGASON covariance of the absolute humidity. Thus, 2 m and surface saturated specific humidity obtained from



the PROMICE AWS are converted to absolute humidity. $w'T'$ is the EC-IRGASON covariance and $T(z)$ and $T_s$ are the $2\,\mathrm{m}$ and surface temperature from the PROMICE AWS. Similar to $u_*$, the $a_*$ and $T_*$ are turbulent humidity and temperature scales. The 10-minute EC-IRGASON data is averaged to hourly values, to match the PROMICE time resolution. The value for the roughness lengths is then determined by filtering the values for neutral conditions ($-0.02 < z/L < 0.02$) and for $z_{0,m}$ for sufficient wind ($U > 3\,\mathrm{m\,s^{-1}}$) and taking the median. The median is taken due to the large range of magnitude in the results,

making the mean unsuitable (see Fig. A1). This gives the following values: $z_{0,m} = 1.3 \times 10^{-4}\,\mathrm{m}$, $z_{0,q} = 5.7 \times 10^{-7}\,\mathrm{m}$ and $z_{0,t} = 2.9 \times 10^{-4}\,\mathrm{m}$. The value for $z_{0,m}$ is similar to the roughness lengths of $1.6 \times 10^{-4}\,\mathrm{m}$ recorded by Van den Broeke et al. (2005b) in the katabatic wind zone in Antarctica.

### 3.3 Model data processing

RACMO2.3 and RACMO2.4 provide the windspeeds at $10\,\mathrm{m}$ height and in the middle of the lowest layer (also at approximately
$10\,\mathrm{m}$ height), respectively. The windspeeds are interpolated to $2\,\mathrm{m}$ height, using the logarithmic wind speed profile and the stability correction functions following the sign convention in Paulson (1970). The surface specific humidity is derived from the surface temperature following the equations in Murphy and Koop (2005), using the equilibrium vapour pressure over ice and assuming saturated conditions.

## 4 Results

### 4.1 Eddy-Covariance flux comparison

The comparison of the LHF measured by the three EC systems over the period 28 May, 2019 to 31 July, 2019 is presented in Fig. 2. The 8-day time series (Fig. 2a-d) illustrate the diurnal fluctuation of the LHF and the dependence of LHF on specific humidity. Besides following the same general trend and diurnal cycle, Fig. 2 confirms that the three EC systems measure similar sub-hourly LHF variations. Hence, we assume that these observed LHF variations are accurate representations of
actual LHF with high signal to noise ratio. Direct comparison of the three instruments (Fig. 2e and 2f) demonstrates high levels of correspondence with correlation coefficient $r = 0.97$ to 0.98, an absolute bias, indicating the mean difference, of $0.2\,\mathrm{W\,m^{-2}}$, and an $RMSE$ between 1.2 and $1.5\,\mathrm{W\,m^{-2}}$. However, the slope of 1.16 in Fig. 2e indicates a substantial difference in the LHF measured by the EC-Li-7500 compared to the EC-IRGASON and EC-KH20 (not shown). This suggests that though the EC-Li-7500 measures the equivalent mean flux as the other two EC systems, it slightly overestimates the amplitude of the daily
LHF.

The difference in the LHF measured by the EC-Li-7500 and the two other EC systems can be explained by the humidity measured by the Li-7500 as shown in Fig. 3. A comparison with the PROMICE AWS (Fig. 3), shows an offset in specific humidity between the PROMICE hygrometer and Li-7500. Apart from this offset, a drift over time in the humidity measured by the Li-7500 can clearly be identified (Fig. 3b). This indicates an incorrect and varying sensitivity of the Li-7500, which
explains the overestimated LHF measured by the Li-7500 and is likely due to an instrumental problem. In an EC-comparison



**Figure 2.** Examples of time series of the LHF measured by (a) the EC-IRGASON, (b) EC-Li-7500 and (c) EC-KH20 and (d) the specific humidity at 2 m and at the surface based on PROMICE AWS measurements. Direct comparisons of the LHF during the entire campaign measured by the EC-Li-7500 and EC-KH20 versus EC-IRGASON are shown in panels e and f. The EC and AWS data have a 10-minute and hourly temporal resolution, respectively.





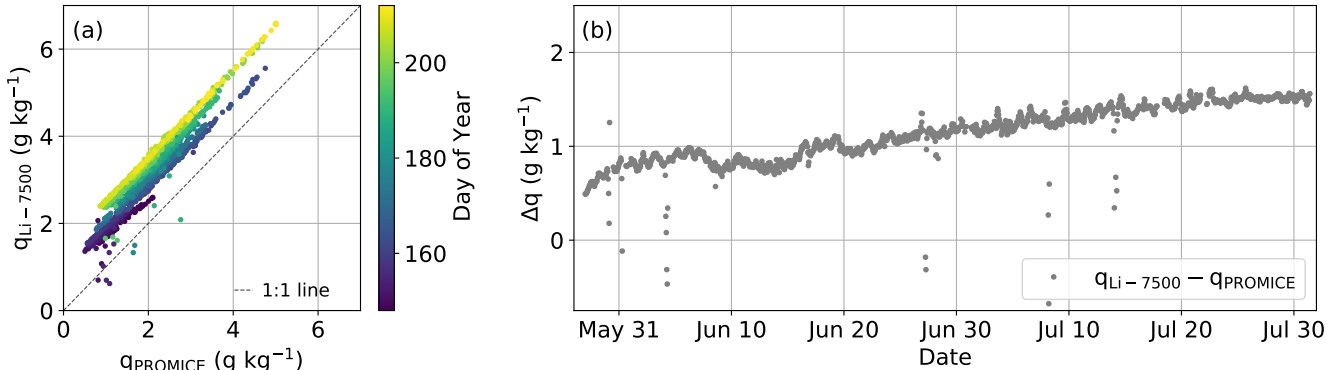

**Figure 3.** (a) Comparison of the hourly averaged specific humidity measured by Li-7500 and PROMICE AWS. (b) The difference between the specific humidity measured by Li-7500 and PROMICE AWS as function of time through the campaign.

study by Polonik et al. (2019) in California, in which a.o. an IRGASON and Li-7500A are compared, a similar instrumental problem is reported. In their study the Li-7500A measured $14\,\%$ higher humidity variances than the IRGASON, resulting in a 10 - $28\,\%$ higher LHF than the IRGASON. In their study, the issue was resolved after instrumental repair and recalibration.

A comparison of the SHF measured by the three EC systems (see Fig. 4) shows even higher agreement between the EC systems than the LHF. The SHFs measured by the three EC systems again follow the same diurnal cycle and have similar sub-hourly variations. The direct comparison of the fluxes also indicates a very good agreement between measurement systems ($r = 0.98$, an absolute bias of less than $0.5\,\mathrm{W\,m^{-2}}$, an $RMSE$ between 1.6 and $1.9\,\mathrm{W\,m^{-2}}$, and slopes of 1.0).

## 4.2 Bulk-method flux validation

Estimates of the LHF and SHF based on the bulk method in the PROMICE data product are overestimated (Fig. 5), as can be seen by the slopes greater than 1 (Fig. 5a and 5e) and the overestimated amplitude in Fig. 5d and 5h. The overestimated flux magnitudes indicate that the roughness length $z_{0,m} = 1 \times 10^{-3}\,\mathrm{m}$ used in the assessed time period is too high for the EastGRIP location. Using the same $z_{0,m}$ for both the LHF and SHF and assuming $z_{0,q} = z_{0,t}$, leads to a larger overestimation of the LHF than the SHF, with a slope of 2.22 as compared to 1.35 (Fig. 5a and 5e). A separate analysis of only the positive and negative fluxes results in similar slope values (Supplementary Material S3), but similar to Box and Steffen (2001) the bulk deposition is slightly more overestimated than bulk sublimation. Using a similar parameterisation (Andreas, 1987) for $z_{0,t}$, $z_{0,m} = 1.3 \times 10^{-4}\,\mathrm{m}$ based on our EC measurements, and assuming $z_{0,q} = z_{0,t}$, improves the correspondence with the EC data (Fig. 5b and 5f), especially the SHF. However, a slightly overestimated LHF remains (slope = 1.44 and absolute bias = $2.44\,\mathrm{W\,m^{-2}}$). Using the parameterisation by Andreas (1987) no other $z_{0,m}$ value could be found that provides good fits between both calculated SHF and LHF bulk fluxes and the EC data. Therefore, we use $z_{0,m} = 1.3 \times 10^{-4}\,\mathrm{m}$ and $z_{0,q} = 5.7^{-7}\,\mathrm{m}$ and $z_{0,t} = 2.9 \times 10^{-4}\,\mathrm{m}$, derived from the EC measurements (see Sect. 3.2). We note that $z_{0,t}$ is larger than $z_{0,m}$ and that $z_{0,t}$ is two orders of magnitude larger than $z_{0,q}$. This is not consistent with previous findings where $z_{0,q}$ and $z_{0,t}$ are often assumed to





**Figure 4.** Examples of time series of the SHF measured by (a) the EC-IRGASON, (b) EC-Li-7500 and (c) EC-KH20 and (d) the temperature at 2 m and at the surface based on PROMICE AWS measurements. Direct comparisons of the SHF during the entire campaign measured by the EC-Li-7500 and EC-KH20 versus EC-IRGASON are shown in panels e and f. The EC and AWS data have a 10-minute and hourly temporal resolution, respectively.

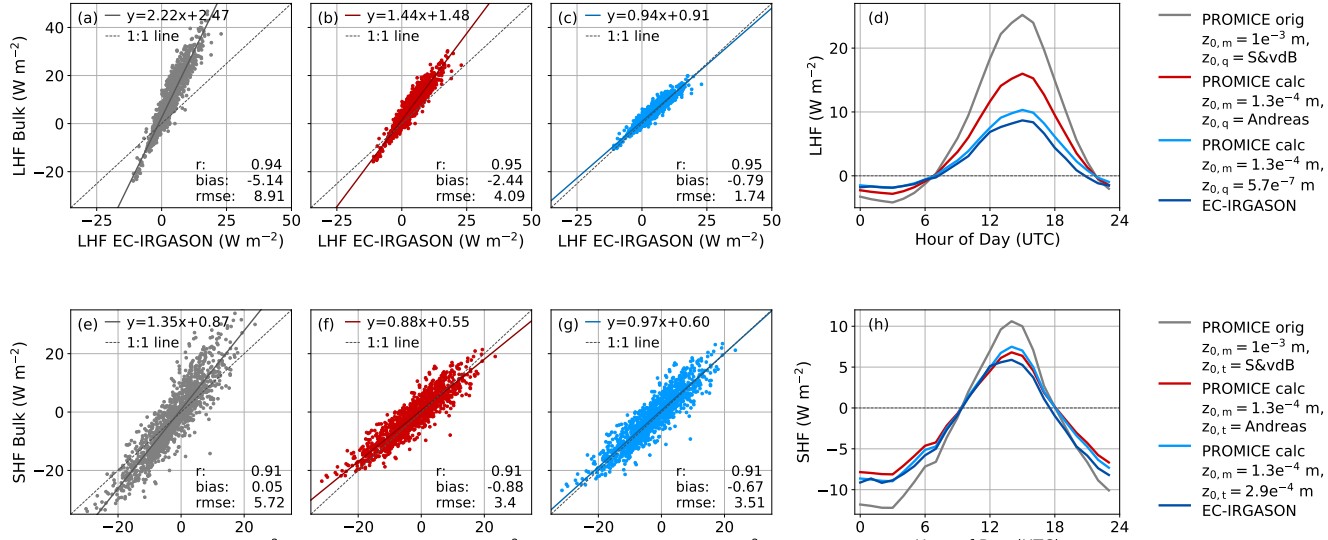

**Figure 5.** Comparison of the bulk flux based on the PROMICE AWS observations and EC-Irgason with the SHF in the top row and the LHF in the bottom row. Panels a and e are based on the original PROMICE data product using $z_{0,m} = 1 \times 10^{-3}$ m and $z_{0,q} = z_{0,t} =$ Smeets and Van den Broeke (2008a, b)(S&vdB). Panels b and f are the bulk fluxes recalculated using the $z_{0,m}$ from the EC-Irgason and parameterizations from Andreas (1987) shown against the observed SHF and LHF from the EC-Irgason. Panels c and d show the calculated bulk fluxes using the $z_{0,m}$, $z_{0,q}$, and $z_{0,t}$ values derived from the EC-Irgason. Panels d and h show the average diurnal cycle of the different flux calculations during the measurement campaign (28 May, 2019 to 31 July, 2019). A time series comparison of the different flux estimates is available in the Supplementary Material S4 (Fig. S3 and Fig. S4).

be smaller or approximately the same as $z_{0,m}$, and $z_{0,q}$ and $z_{0,t}$ having the same or nearly identical values (Van Tiggelen et al., 2023; Andreas, 1987; Smeets and Van den Broeke, 2008a). Using the above roughness lengths, we find that the bulk method provides similar LHF and SHF values as the EC-Irgason (absolute bias of $0.79\,\mathrm{W\,m^{-2}}$ and $0.67\,\mathrm{W\,m^{-2}}$, respectively, Fig. 5c

and 5g) and these values are therefore used in the rest of this study and assumed constant. It is expected that the roughness length of a snow surface varies seasonally, e.g. through higher winds in winter that promote surface snow structure change (Zuhr et al., 2021). However, as no EC measurements were conducted during the winter, we only evaluate the estimated bulk fluxes during the summer (daylight) period. We assume that using the improved roughness lengths the calculated bulk fluxes provide reliable estimates during the winter as well.

**4.3 Model comparison**

Intercomparisons between the simulated LHF from the RACMO and MAR models and observations are shown in Fig. 6. We note that there is a large spread between models and the observations in the seasonal cycle (Fig. 6a). The monthly averaged bulk LHF from the AWS varies between $2.6\,\mathrm{W\,m^{-2}}$ and $-0.9\,\mathrm{W\,m^{-2}}$. The difference between the monthly averaged models



and observations is up to $2.1\,\mathrm{W\,m^{-2}}$ and $-1.7\,\mathrm{W\,m^{-2}}$ in June and October, respectively. Except for RACMO2.3 in summer and RACMO2.4 in June, the RCMs underestimate the LHFs year-round. Figure 6b shows the cumulative sum of the LHF over the 3.5 years of our measurements, expressed as a mass flux. While the cumulative mass flux from the AWS is approximately net zero after three complete years, simulations using MAR and RACMO2.4 result in a net deposition, with a total difference of $25\,\mathrm{mm\,w.e.}$ and $38\,\mathrm{mm\,w.e.}$, respectively. RACMO2.3 shows overestimated sublimation and a total of $-5\,\mathrm{mm\,w.e.}$ by the end of the three years. The time series in Fig. 6c shows that both the models and observations have similar diurnal cycles of the LHF in summer, although the amplitudes differ. In winter (Fig. 6d) there is a large difference between the models and the observations. MAR and RACMO2.4 episodically show strongly negative fluxes (of $-7\,\mathrm{W\,m^{-2}}$ to $-13\,\mathrm{W\,m^{-2}}$), while the bulk flux from the AWS remains close to zero (between $1\,\mathrm{W\,m^{-2}}$ and $-2\,\mathrm{W\,m^{-2}}$). RACMO2.3 shows partly opposite flux results of MAR and RACMO2.4 and simulates a positive LHF, whereas the other two simulate a negative flux.

A direct comparison between model output and observed LHF is complicated by the sublimation of drifting snow, which, when it occurs, varies strongly with height and can reach significant heights above the surface (Palm et al., 2018). In the observations presented here, by locally moistening and cooling the air, blowing snow sublimation would violate the constant flux assumption in the EC measurements and bulk flux calculations. RACMO2.3 has an idealised drifting snow scheme (Lenaerts et al., 2010), in which the associated vertically integrated sublimation is fully added to the surface latent heat flux, potentially explaining the overestimated sublimation in this model. In RACMO2.4 the drifting snow scheme is improved and moisture from drifting snow sublimation is added directly to the associated atmospheric model layers (Gadde and Van de Berg, 2024). Given the strong vertical variations in drifting snow sublimation, here we only use the surface latent heat flux from RACMO2.4 for comparison to observations, which may lead to underestimated sublimation. In MAR, the drifting snow routine was not activated for this comparison.

The comparison of SHF from observations, the bulk method and the RCMs is shown in Fig. 7. When comparing the seasonal cycle based on monthly means (Fig. 7a), the models have a slightly lower to similar SHF compared to the observations in summer, but simulate considerably more negative SHF values in winter. A monthly difference between the models and observations in winter is between $-20$ to $-33\,\mathrm{W\,m^{-2}}$ (Fig. 7a). The time series in Fig. 7b and 7c show a similar pattern as the LHF comparison. During the summer the models and observations show similar diurnal cycles but differ in amplitude and hour-to-hour variations (Fig. 7b). In winter a large difference between the models and observations can be seen, with strongly negative hourly fluxes of up to $-62\,\mathrm{W\,m^{-2}}$ to $-86\,\mathrm{W\,m^{-2}}$ from the models and a flux close to zero ($-2\,\mathrm{W\,m^{-2}}$ to $1\,\mathrm{W\,m^{-2}}$) from the estimates from the bulk method applied to AWS data (Fig. 7c).

## 5 Discussion

### 5.1 Eddy-Covariance intercomparison

We compare our results from the three EC systems in the polar conditions of the Greenland Ice Sheet to the performance of similar EC systems under non-polar conditions from earlier studies. In Schmidt et al. (2012) a verified and calibrated EC system was used to validate 84 EC systems of the AmeriFlux network, which are spread out over the North American continent,





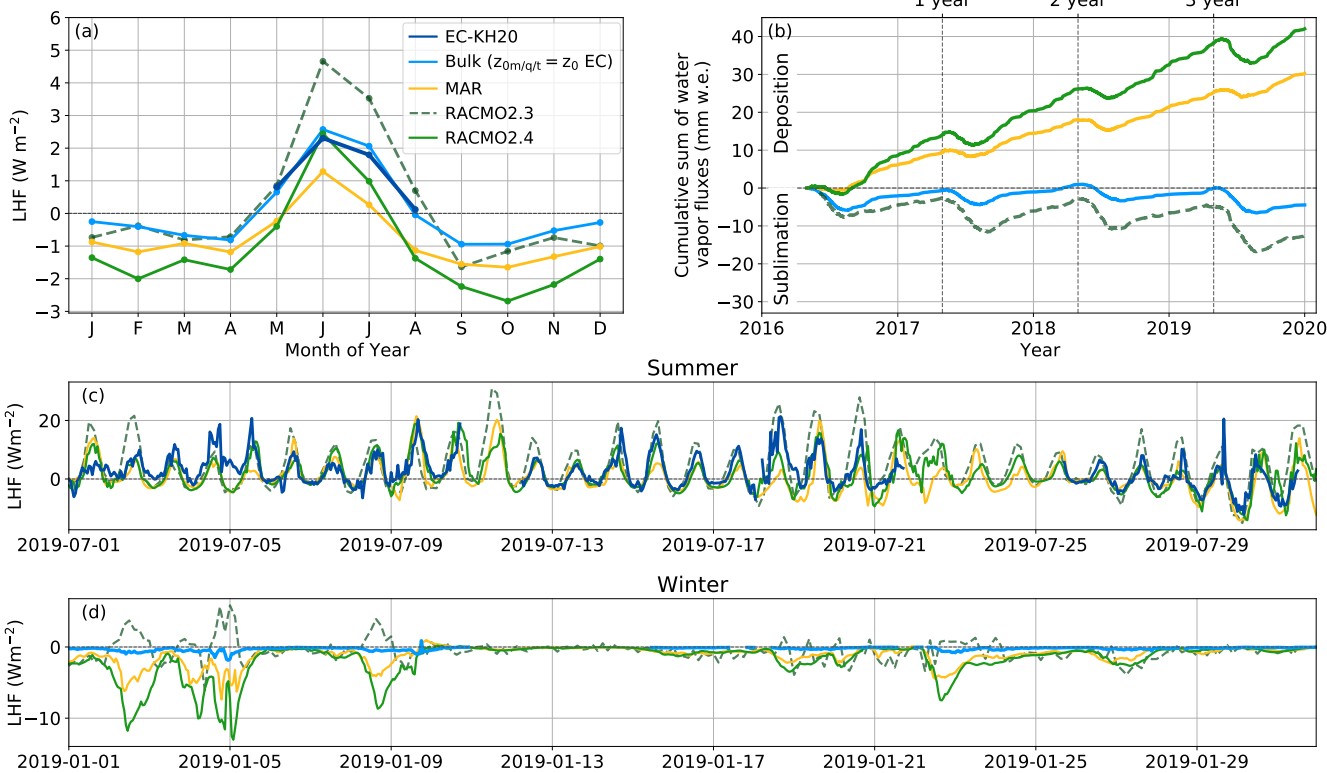

**Figure 6.** (a) Seasonal cycle of the LHF from May 2016 until the end of 2019. (b) Cumulative LHF expressed in mm water equivalent (mm w.e.) (note the fluxes follow the SMB sign convention in this panel). Panels c and d show examples of time series of the LHF during summer and winter, respectively. LHFs are shown based on observations from the EC-KH20 and from calculations using the bulk method using observations from the PROMICE AWS. Simulated LHFs are based on the RCM's MAR, RACMO2.3 and RACMO2.4. The observational time series from the EC-KH20 is used due to its longer available record.

hence covering a wide range of environments. They reported a relative instrumental error of $3.06 \pm 15.60\,\%$ for the SHF and $1.72 \pm 10.58\,\%$ for the LHF, for the systems using an open path gas analyser. Polonik et al. (2019) also compared different EC systems, using a combination of a CSAT3A, Gill 3R-50, IRGASON and Li-7500 A, in California. For the SHF, they generally

find a slope between 0.92 and 1.09, with an intercept between $-0.22$ and $0.92\,\mathrm{W\,m^{-2}}$ (in one case, they report an intercept of $7.17\,\mathrm{W\,m^{-2}}$) and for the LHF a slope between 0.96 and 1.07 and an intercept between $-0.2\,\mathrm{W\,m^{-2}}$ and $-0.01\,\mathrm{W\,m^{-2}}$. Mauder and Zeeman (2018) compared six different sonic anemometers, CSAT3, Gill HS-50 and R3, METEK uSonic-3 Omni, R. M. Young 81000 and 81000RE, in southern Germany. Good agreement was found for the SHF with a slope between 0.98 and 1.02 and an intercept between 1.2 and $-2.5\,\mathrm{W\,m^{-2}}$. A comparison study by Wang et al. (2016), was carried out in a

cold desert environment in northwestern China using an IRGASON and a Windmaster Pro. They found a slope of 1.11 and an intercept of $1.22\,\mathrm{W\,m^{-2}}$ for the SHF. Finally, a comparison study by Loescher et al. (2005) was done in Oregon, comparing





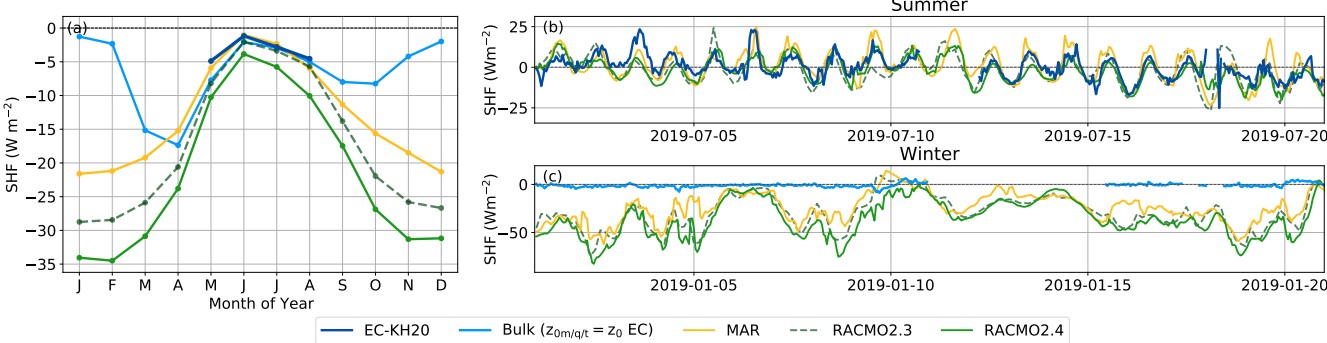

**Figure 7.** (a) Seasonal cycle of the SHF from May 2016 until the end of 2019. Panel b and c examples of SHF time series for a 21-day period in summer and winter, respectively. SHFs are shown based on observations from the EC-KH20 and from calculations using the bulk method using observations from the PROMICE AWS. Simulated SHFs are based on the RCM's MAR, RACMO2.3 and RACMO2.4. The observational time series from the EC-KH20 is used due to its longer available record.

eight different sonic anemometers (a-Probe, k-Probe, CSAT3, R3, DA-600 (TR61A), SAT-550, USA-1 and RM81000), where they found $-1$ to $8\,\%$ difference in the SHF.

We find an inter-instrument variability for the SHF with $r = 0.98$, slope $= 1.0$, intercept $= 0.4\,\mathrm{W\,m^{-2}}$, bias between $-0.4$
and $-0.5\,\mathrm{W\,m^{-2}}$ and an $RMSE$ between 1.6 and $1.9\,\mathrm{W\,m^{-2}}$. For the LHF is $r$ between 0.97 and 0.98, slope between 1.01 and 1.16, intercept between $-0.1$ and $-0.2\,\mathrm{W\,m^{-2}}$, bias between 0.2 and $-0.2\,\mathrm{W\,m^{-2}}$ and an $RMSE$ between 1.2 and $1.5\,\mathrm{W\,m^{-2}}$. This inter-instrumental variability is consistent with the above-mentioned campaigns in non-polar conditions. Hence, we conclude that the accuracy and precision of EC SHF and LHF measurements carried out on the top of the Greenland Ice Sheet are comparable to EC measurements carried out under less difficult conditions. Our measurements and their high
level of correspondence provide confidence in the quality of EC systems deployed under harsh polar conditions. However, systematic biases with the EC method, due to boundary layer characteristics in polar conditions cannot be ruled out.

### 5.2 Eddy-Covariance and Bulk Flux Comparison

Our comparison between the EC measurements and the SHF and LHF deduced from the bulk method confirms the generally accepted uncertainties and challenges that come with using the bulk method in polar conditions. Known major limitations of
370 MOST are the underpinning assumptions, i.e. high Reynolds number flow, steady state conditions and flat homogeneous terrain, in combination with the uncertainties in the universal functions describing the non-dimensional flux-gradient relationships (Foken, 2006; Högström, 1988). In this study, we also highlight the strong dependence of the calculated fluxes on the value of $z_{0,m}$ and the parameterisation used to obtain $z_{0,q}$ and $z_{0,t}$. Different roughness lengths and parameterisations can lead to strong over- or under-estimation of the diurnal flux amplitude which has large consequences for the calculated net and total fluxes. Our
results underline that uncertainty from using the bulk method can be greatly reduced by having a period of combined AWS and EC measurements in which not only $z_{0,m}$ is determined but also the fluxes are directly compared. Further improvement of the





accuracy of the bulk method might be achieved by also having EC measurement during winter, making year-round evaluation of the roughness length possible.

The two orders of magnitude difference between the derived $z_{0,t}$ and $z_{0,q}$ explain the different accuracy of the bulk LHF and SHF when it is assumed that $z_{0,q} = z_{0,t}$ (Fig. 5abef). Comparing a filtered selection of all the measured $z_{0,t}$ and $z_{0,q}$ values during our campaign with different parameterizations shows that the measured $z_{0,t}$ is consistently higher than the parameterised $z_{0,t}$ and the measured $z_{0,q}$ is consistently lower than $z_{0,t}$ (Appendix A, Fig. A2). We note that the calculation of the roughness lengths is based on MOST and the accuracy of the temperature and humidity gradient measured by the PROMICE AWS. Our result implies that the parameterisation for $z_{0,t}$ and the common assumption that $z_{0,t}$ is equal to $z_{0,q}$ used in most models and datasets, might not be valid over the entire Greenland Ice Sheet.

## 5.3  PROMICE data during winter

There is a large difference between the wintertime SHF/LHF values simulated by the RCMs and the bulk flux calculations (Fig. 6 and Fig. 7). It is not possible to reconcile those discrepancies within the measurement uncertainties and hence this indicates either large systematic biases in models, issues with the measurements, assumptions related to the measurements or a combination of those. Intercomparison of the independent $2\,\mathrm{m}$ temperature and wind speed measured by the PROMICE and GC-Net AWS show similar characteristics. During Arctic winter, conditions are mostly quasi-saturated so that the specific humidity predominantly depends on air temperature, and the uncertainty in the relative humidity is unlikely to have a significant impact on the fluxes. The most important observational uncertainty is therefore that of the surface temperature, which is derived from the measured upward longwave radiation, especially since the surface temperature is key for determining the near-surface temperature gradient, which determines the difference between flux results from the model simulation and observations (see Sect. 5.4 and Fig. 9). We note that the net longwave radiation is close to zero for long periods spanning several days to over a week (Fig. 9a). The net zero longwave radiation sometimes coincides with low temperatures ($T < -50\,^\circ\mathrm{C}$), conditions normally associated with a surface based temperature inversion rooted in longwave radiative cooling (Van den Broeke et al., 2004; Miller et al., 2017). A zero net longwave radiation could also be explained by frosting of the sensor, which leads to artificial neutral conditions and SHF/LHF values close to zero (Fig. 9b). Measurements from both Summit in Greenland (Miller et al., 2013; Berkelhammer et al., 2016), the katabatic wind zone in Antarctica (Van den Broeke et al., 2005a, 2009) and modelling studies (Shahi et al., 2020) show that neutral conditions are uncommon during winter time. To explore whether frosting obstructs the radiation sensor we compare the near-surface temperature gradient estimated from the PROMICE AWS with the near-surface air temperature gradient obtained from the two independent GC-NET AWS temperature measurements installed at two different heights (Fig. 8e and 8f). One can notice two important findings: 1) The independently observed temperature gradients by the single-level PROMICE and two-level GC-NET AWS are generally approximately similar and an order of magnitude smaller than the modelled temperature gradient (see Fig. 9i). 2) Features in the synoptic-scale variability of the near-surface temperature gradient are observable in both AWS datasets.

A second line of evidence in support of the radiation sensor measurements can be deduced from the shallow ($10\,\mathrm{cm}$) snow temperature, which is less prone to measurement uncertainties (Fig. 8c and 8d). One would not expect the magnitude of the





temperature gradient deduced from the near-surface snowpack temperature to be equivalent to the temperature gradient deduced from the snow skin surface temperature due to the thermal mass of the snow. However, we note the general good agreement in the evolution of the sign of the temperature gradient based on the snowpack and skin surface temperature. For example, when colder air is present, the air temperature is lower than the snow surface (e.g. 1st until the 10th of January 2019) and vice-versa (e.g. 10th until 15th of January). Interpretation is, however, not straightforward, as the thermal inertia of the snowpack leads to a delayed response to surface forcing, and snow temperature reflects a delayed and smoothened pattern of the air temperature, reflecting a mixture of heat exchange between air, surface and subsurface at longer time scales.

Altogether, based on our assessment of observations and models, we cannot decide with strong certainty whether modelled or observed wintertime SHF/LHF values are closer to reality, and any conclusions based on either data source should be drawn with care. The large model-data discrepancy and our discussion here illustrate the outstanding requirement to resolve this question for areas with wintertime conditions such as the EastGRIP location and the need for more direct observations of the near-surface temperature gradient.

## 5.4  Observations and models comparison

Assuming that observational uncertainty cannot fully explain the difference with the modelled fluxes during winter, we explore potential shortcomings in the model simulations of either the air temperature or the snow surface temperature. The time series in Fig. 9 of the LHF, SHF and driving variables show that, while the wind speed in all three models is similar to the observed wind speed, both the humidity and temperature gradient are much larger than the observations. This means that the larger modelled magnitudes of LHF and SHF in winter result from a larger near-surface temperature gradient and therefore also a larger humidity gradient in the models. Since EastGRIP is located centrally in the modelling domain of both MAR and RACMO, we assume the impact of ERA5 forcing at the lateral boundaries to be small. With the tentative assumption that the observed surface temperatures are correct, the evaluated climate models simulate a too strong stability in this part of the ice sheet during winter. This implies that the atmospheric processes driving the surface gradients are not yet accurately modelled and parameterised. This persistently larger near-surface temperature gradient during winter can have a considerable impact, as the LHF and SHF are important for obtaining an accurate SMB and closing the SEB. As seen in Fig. 6b small differences in LHF estimates can lead to large differences on longer timescales. Although studies suggest that the contribution of the LHF might switch to net mass loss in the future (Cullen et al., 2014), most RCMs simulate the LHF to be a positive contributor to the SMB to date. In fact, by combining the MAR model with summer observations at EastGRIP, Dietrich et al. (2024) find the LHF to be a negative SMB contributor in their simulations. Similar to Dietrich et al. (2024), we find that RACMO in winter also overestimates deposition compared to AWS observations at EastGRIP. This raises the important question of whether both established RCMs face systematic errors in the representation of the surface boundary layer on ice sheets during winter. Besides their importance for obtaining an accurate SEB and SMB, which are used for sea level rise estimates, accurately simulating surface fluxes is also important for other fields of study such as the interpretation of water isotope climate proxies from the snowpack and ice cores (Wahl et al., 2022; Dietrich et al., 2023). Underscoring the need for better knowledge and model representation of the surface near-stability during winter on the Greenland Ice Sheet, both by improving model parameterisations and increasing the number



**Figure 8.** Time series of winter AWS observations with (a) the net longwave radiation, (b) the calculated sensible and latent heat flux, using the $z_{0,m}$, $z_{0,q}$ and $z_{0,t}$ values derived from the EC-Irgason, (c) the air temperature at $2.6\,\mathrm{m}$ above the surface and the snow temperature approximately $10\,\mathrm{cm}$ below the surface, (d) the temperature difference between the $2.6\,\mathrm{m}$ and snow temperature and (e) the PROMICE temperature difference between $2.6\,\mathrm{m}$ and the surface, determined via the longwave radiation and (f) the GC-Net temperature difference between $2.3\,\mathrm{m}$ and $1.1\,\mathrm{m}$. Note that the complete dataset is shown for this figure, including the data below $-50\,^{\circ}\mathrm{C}$. Comparison of the PROMICE AWS and model temperature gradients is analysed in Fig. 9. Separate up- and downward longwave radiation and a magnified view of the near-zero variability of the net longwave radiation are provided in Supplementary Material S5.





**Figure 9.** Examples of winter time series showing (a) the SHF, (c) LHF, (e) windspeed, (g) specific humidity difference between $2\,\mathrm{m}$ and surface level and (i) temperature difference between $2\,\mathrm{m}$ and surface level. Panels b, d, f, h and j show boxplots over the same period, where the whiskers indicate the 5th-95th percentile, the box the 25th to 50th percentile, the thick line the median and the black dash (-) the mean.

of observations. This includes obtaining direct observations of the surface temperature, but also observations of blowing snow, to better understand and parameterize the impact of blowing snow sublimation on the heat fluxes.

# 6 Conclusions

An instrument intercomparison of three co-located EC systems in the interior of the Greenland Ice Sheet shows that the EC method provides accurate LHF and SHF measurements during summer on the ice sheet. This is based on the high levels of 450 correspondence of the measured fluxes by the three EC systems. Differences in the fluxes measured by the different systems





can be explained by instrumental error, as is the case for the LHF measured by the EC-Li-7500 in our study, or otherwise, fall
into the uncertainty range documented by other EC comparison studies in non-polar conditions. Comparing the validated EC
fluxes with fluxes obtained using the bulk method confirms the uncertainty of the bulk method. Besides already known issues,
like limitations of MOST, this study highlights the dependence of the flux calculation on the value of $z_{0,m}$ and the limitation
of parameterisations for obtaining $z_{0,q}$ and $z_{0,t}$. Specifically, using a fixed roughness length over the ice sheet introduces a
large bias in the bulk flux measurement compared to EC measurements, and the often-used assumption that $z_{0,q}$ and $z_{0,t}$ are
equal does not seem to hold in this location. Therefore, this study highlights the need for more EC measurements in the centre
of the ice sheet, to improve both existing bulk flux estimates and roughness length parameterisations. Lastly, comparison of
observations with RCMs MAR, RACMO2.3 and RACMO2.4 shows large differences between the simulated and observed
LHF and SHF, especially during winter. Our results indicate that the winter difference comes from a consistent overestimation
of atmospheric stability at EastGRIP in climate models, but observational uncertainty due to frosting cannot be ruled out and
further measurements are needed to support these results. Too strong near-surface gradients during winter in the models result
in persistently larger turbulent exchange, leading to errors in the contribution of the LHF to the SMB of the ice sheet and both
the LHF and SHF to the SEB. Contemporary and future SMB estimates based solely on models might thus be less certain than
465 previously thought, underlining the need for improving model parameterisations and obtaining more and reliable observations
on the Greenland Ice Sheet in particular and polar regions in general.

*Data availability.* The EC-KH20, EC-IRGASON and EC-Li-7500 data from the summer of 2019 will be uploaded to Pangaea during the
review process. EC-KH20 data from the summers of 2016, 2017, 2018 is available on Pangea (https://doi.pangaea.de/10.1594/PANGAEA.
962310, Steen-Larsen et al. (a), https://doi.pangaea.de/10.1594/PANGAEA.962311, Steen-Larsen et al. (b), https://doi.org/10.1594/PANGAEA.
946741, Steen-Larsen et al. (2022)). The PROMICE AWS product is available at https://doi.org/10.22008/FK2/IW73UU (How et al., 2022).
The GC-Net data is available at https://doi.org/10.22008/FK2/VVXGUT (Steffen et al., 2022). The MAR simulations are available on Zen-
odo (https://doi.org/10.5281/zenodo.8335402, Dietrich (2023)). The three-hourly RACMO2.3p2 and hourly RACMO2.4p1 time series from
2010 to 2020 from the grid cell closest to EastGRIP will be uploaded to Zenodo during the review process.

**Appendix A**

475 An overview of the roughness length values for momentum, moisture and heat measured during the campaign, filtered for
neutral conditions (-0.02 $< z/L <$ 0.02) and for $z_{0,m}$ for sufficient wind ($U >$ 3 m s$^{-1}$), is shown in Fig. A1.

A comparison of the roughness length values obtained from the EC-IRGASON and different roughness length parameteri-
sations is provided in Fig. A2. To limit noise, the following selection is applied to the roughness length values displayed in Fig.
A2: $U >$ 3 m s$^{-1}$, $|q_{2m} - q_{surf}| >$ 0.1 g kg$^{-1}$, $|t_{2m} - t_{surf}| >$ 0.5 °C, $|z/L| <$ 0.2, $1 < \sqrt{w}/u_* < 1.5$ and
480 0.1 m s$^{-1} < |u_*| <$ 1.5 m s$^{-1}$.

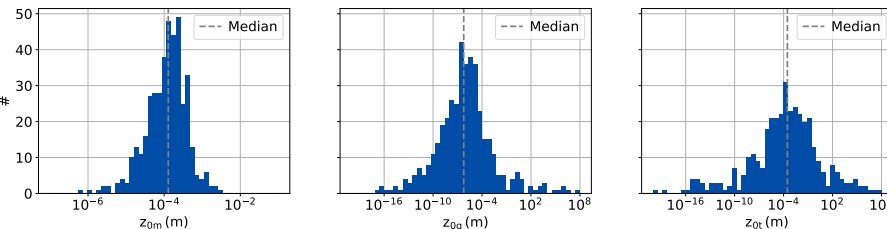

**Figure A1.** Histograms of the hourly roughness length values derived from the EC-IRGASON and PROMICE AWS observations.

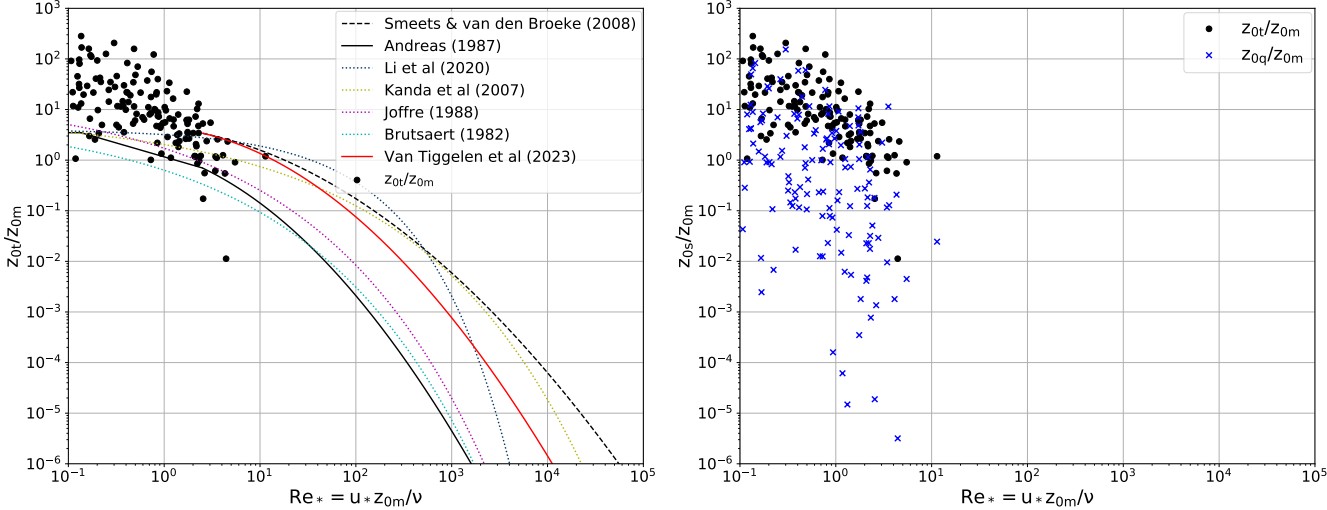

**Figure A2.** The hourly ratio of $z_{0,t}/z_{0,m}$ from the EC-IRGASON and PROMICE AWS observations against the roughness Reynolds number $Re_* = u_* z_{0,m}/v$. Plotted together with (a) lines denoting the parameterisation from Smeets and Van den Broeke (2008b), Andreas (1987), Li et al. (2020), Kanda et al. (2007), Joffre (1988), Brutsaert (1982) and Van Tiggelen et al. (2023) and (b) the hourly $z_{0,q}/z_{0,m}$ ratio from the EC-IRGASON and PROMICE AWS observations. The EC-IRGASON values are hourly averages of the 10-minute covariances.

*Author contributions.* IH wrote the initial draft with contributions from all authors. The study conceptualization were carried out by HCSL, JR, SW, and IH. Formal analysis was carried out by IH and LJD with support from STK, MvT, SW, and HCSL. Methodology was developed by HCSL and SW. IH was supervised by HCSL, JR, MvdB, and MvT. Model simulations were provided by LJD, MvdB, and MvT. Instrumental resources were provided by JR, AH, JEB. Funding acquisition and project administration was done by HCSL.

*Competing interests.* At least one of the (co-)authors is a member of the editorial board of The Cryosphere.



*Financial support.* This research has been supported by the European Research Council, the European Union's H2020 program (SNOWISO, grant no. 759526)

*Acknowledgements.* This paper is a contribution to the H2020 European Research Council Starting Grant project SNOWISO (grant agreement no. 759526). EGRIP is directed and organized by the Centre for Ice and Climate at the Niels Bohr Institute, University of Copenhagen.
It is supported by funding agencies and institutions in Denmark (A. P. Møller Foundation, University of Copenhagen), USA (US National Science Foundation, Office of Polar Programs), Germany (Alfred Wegener Institute, Helmholtz Centre for Polar and Marine Research), Japan (National Institute of Polar Research and Arctic Challenge for Sustainability), Norway (University of Bergen and Trond Mohn Foundation), Switzerland (Swiss National Science Foundation), France (French Polar Institute Paul-Emile Victor, Institute for Geosciences and Environmental research), Canada (University of Manitoba) and China (Chinese Academy of Sciences and Beijing Normal University). Data from the Programme for Monitoring of the Greenland Ice Sheet (PROMICE) are provided by the Geological Survey of Denmark and Greenland (GEUS) at http://www.promice.dk. Data analysis and model simulations were performed on resources provided by Sigma2 – the National Infrastructure for High Performance Computing and Data Storage in Norway. AH acknowledges funding from UArctic Chair and from the Research Council of Norway (332635 & 342265), the Research Council of Finland (363970), and the Fulbright Commission. We want to thank Christiaan van Dalum for proving the data from RACMO2.4p1. We want to thank Alexandra Zuhr and Hanna Meyer for servicing the EC systems during the field campaign.



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
