# Peer review of "On the accuracy of the measured and modelled surface latent and sensible heat flux in the interior of the Greenland Ice Sheet"

_EGUsphere, 2025_

## Author Comment (AC1)

Reply to RC1: 'Comment on egusphere-2025-711', Anonymous Referee #1, 09 Apr 2025. In the following text, the Referee's comments are reported in bold text and the author's answers are noted in italics. The edited text in the manuscript is in green.

We thank the reviewer for the positive and constructive feedback, and we agree with the summary provided by the reviewer. We believe we have addressed all the comments and will go through them step by step below.

**The present study proposes an analysis of observations of sensible and turbulent heat fluxes during a few months in 2019 within the Greenland Ice Sheet, from three different eddy covariance measurement systems. This is a unique dataset, extremely difficult to obtain, and allows us to explore the importance of latent heat flux on the local mass balance. After presenting an intercomparison between the three measurement systems (which ultimately agree very well), the data are used to train a 'bulk transfer' type relationship in winter and in other years, enabling comparison with turbulent fluxes simulated by two climate models. The multiple challenges of measuring and simulating these fluxes in this type of environment are then discussed.**

**I really enjoyed reading this paper. The plots are clean and well put together. The text is well written and the sequence of ideas is easy to follow. With the modifications I recommend below, I believe this article has a rightful place in a journal of the caliber of The Cryosphere.**

**GENERAL COMMENTS**

**LHF and SHF in winter**

**I am puzzled by the 'observed' winter fluxes. The approach used to estimate these fluxes is to apply bulk transfer from a standard weather station based on z0m, z0t and z0q values calibrated in summer. It is clear to me that these values do not hold in winter, partly because the surface does not have the same roughness. More troubling is that the measured temperature difference between the surface and the air about 2.6 m above is very small in winter (at most 1°C in Jan. 2019!!). This seems impossible. Since qsfc depends on Tsfc, both turbulent fluxes are affected.**
**I know that most of these issues are raised by authors. I would like to see more options explored to improve these winter results:**
*We agree with the reviewer that the winter temperature gradients are puzzling, but we want to highlight that this very small temperature gradient is measured by both the PROMICE and GC-Net AWS. This, together with the arguments provided in the manuscript, is reason for us to explore the possibility that the small temperature gradient is not a measurement error, but could also indicate some limitation in our understanding of arctic boundary layer processes, e.g. concerning katabatic wind flows, in the winter.*

**- Have you tried estimating winter z0m from standard weather station data or other means?**

*Unfortunately, this is not possible using the available AWS observations alone, since an independent measurement of either the flux (u\*) or the gradient (from two wind speed measurement levels or more) would be needed. Although the GC-Net AWS does provide measurements at two levels, the uncertainty in extrapolating $z_{0,m}$ would be too large, due to the combination of the measurement levels being close together and the small gradients. Although other options for obtaining estimates of the surface roughness using remote sensing exist, e.g. using photogrammetry or laser altimetry (van Tiggelen et al., 2021), radar altimetry (Scanlan et al., 2025) or GNSS reflectometry (Pickell et al., 2025), we believe this falls outside the scope of this manuscript.*

*In order to still get a better understanding of the uncertainty of the fixed roughness length, we have conducted a sensitivity study to investigate the influence of the roughness length on the resulting (winter) fluxes (see review 2). The sensitivity study is added to the supplementary material and expanded on in the manuscript.*

**- Have you explored surface emissivity values other than 0.97? Satellite data for Tsfc?**

*The reason for using a surface emissivity of 0.97 is that this is also used in Fausto et al. (2021). We have added a sensitivity study using different emissivities below. Here the seasonal cycle of the PROMICE near-surface atmospheric temperature gradient ($T_{2m}$ - $T_{surf}$) is shown. In the shaded areas, the same gradient is shown, but with $T_{surf}$ calculated following Fausto et al. (2021) using different emissivity values. Here it can be seen that although using different surface emissivity values leads to some difference in surface temperature and therefore also the temperature gradient, this is limited during winter. So the choice of surface emissivity cannot explain the near-neutral temperature gradients during the winter months.*

$$T_s = \left( \frac{LR_{out} - (1-\epsilon) \cdot LR_{in}}{\epsilon \cdot 5.67 \times 10^{-8}} \right)^{0.25} - 273.15$$

[Figure]

*Remote sensing estimates of the land surface temperature are available through infrared radiation measurements. However, as these measurements are both influenced by atmospheric absorption and also depend on the snow emissivity, the resulting uncertainty is similar to the in-situ observations.*

**- What about radiative flux divergence? See for instance:**
**https://journals.ametsoc.org/view/journals/apme/46/9/jam2542.1.xml**
*We thank the reviewer for this suggestion. However in the paper two radiometers at 2 and 48 m are used. As the PROMICE AWS at EastGRIP only has one radiometer, this would only be possible when the surface temperature is known, meaning when it's melting. As temperatures are below zero most of the year round at EastGRIP, this is unfortunately not possible.*

**Wind**

**While temperature and humidity gradients play a key role in turbulent fluxes, wind is also key. This is barely mentioned in the article. What about katabatic winds at the measurement site? How do they affect the results? This should be covered in the introduction, results and discussion.**
*The reviewer is correct that winds play an important role in turbulent fluxes. However, for the EC-comparison, a windspeed filter is applied to the data to ensure suitable conditions, and in the model comparison, we see a high similarity between the windspeed in the models and observations. Therefore, this does not come back in the discussion as much.*
*With regard to the katabic winds, EastGRIP is indeed in the katabatic wind zone, as can be seen in the wind rose below of the wind speed and direction recorded by the PROMICE AWS in the period from 2016 to 2020. We have added more information in the description of the conditions at EastGRIP and in the discussion of the EC comparison and the model comparison about the katabatic winds and the possible influences.*

[Figure]

*"EastGRIP is located in the north-eastern part of the Greenland Ice Sheet (Fig. 1a), in the katabatic wind zone, and approximately 350 km NNE of Summit, the highest point of the ice sheet."*

*"The mean windspeed was 4.5 m s⁻¹, with a maximum of 13.7 m s⁻¹, an average direction of 254°, and strong directionality due to the katabatic winds (Fig. 1b)."*

*"The average wind direction during this 4-year period was 242°, with 74 % of the time the wind direction falling within the 200 − 280° sector."*

*"However, systematic biases with the EC method, due to boundary layer characteristics in polar conditions cannot be ruled out, such as the influence of the katabatic wind maximum, and during (sub)meso motions (Lan et al., 2022), which both typically take place under very stable conditions and can cause the transport to deviate from a fully turbulence-dominated regime."*

*"They propose that the difference in LHF between the model and observations during summer arises from a negative bias in downwelling longwave radiation, as also found by Fettweis et al. (2017), from the cloud scheme, while the winter bias may be caused by vertical mixing through katabatic winds that is not represented in the model (Dietrich et al., 2024)"*

**SPECIFIC COMMENTS**

**Abstract: Mention the exact dates of the measurement period.**
*The exact dates are added during which the EC intercomparison was carried out, together with the years over which the model comparison is done.*

*"In this study, we present an intercomparison of three independent EC systems from the 28th of May 2019 until the 31st of July 2019 at the EastGRIP site at ~ 2700 m a.s.l on the Greenland Ice Sheet to assess the accuracy of LHF and SHF measurements."*

*"Using improved values for z0,m, z0,q and z0,t, recomputed bulk fluxes are compared to fluxes simulated by regional climate models MAR, RACMO2.3p2 and RACMO2.4p1 for the period from 2016 to 2020."*

**L22: I know this comes up later, but the introduction should distinguish between surface and blowing snow sublimation.**
*We have clarified this sentence, so it is clear the study focuses on surface sublimation.*

*"The turbulent latent heat flux (LHF) directly impacts the SMB by adding or removing water through surface evaporation/sublimation and condensation/deposition and linking the mass balance to the surface energy balance (SEB)."*

**L33: According to your Fig. 5h, a third of the day in summer experiences unstable conditions. I think we tend to assume that as soon as there is snow or ice, the atmosphere is necessarily stable, which is not the case.**
*Because this feature is surprising many, we have added a sentence to the manuscript which highlights the observation:*
*"It is noteworthy that contrary to the commonly assumed stable atmospheric boundary layer over snow surfaces, in 18 %, 21 % and 21 % of the 10-min time intervals measured by the*

*EC-IRGASON, EC-Li-7500 and EC-KH20, respectively, the conditions were unstable (z/L<-0.02)*"

**L43: Discuss the strengths and weaknesses of open-path and closed-path gas analyzers.**
*We have added a sentence on the main weaknesses and strengths of open- and closed-path gas analysers.*

*"Wind and temperature fluctuations are typically measured at high temporal resolution (10 − 50 Hz) using a sonic anemometer, along with humidity variations using an open or closed path water vapour analyser. However, both suffer from limitations as open path analysers experience less attenuation than closed path analysers, but open path analysers are more sensitive to disturbances such as precipitation (Polonik et al., 2019)."*

**L54: you need to elaborate on the limitations of the MOST approach - what exactly is at stake during stable atmospheric conditions?**
*We have added a sentence elaborating on the limitations of MOST.*

"*The existing limitations in the validity of MOST, in particular for moderately and strongly stable atmospheric conditions (e.g., Grachev et al., 2005; Cullen et al., 2007; Schlögl et al., 2017; Pfister et al., 2019), which are common on top of the Greenland Ice Sheet during winter (Cullen and Steffen, 2001), are expected to increase uncertainty for corresponding SHF and LHF bulk estimates. This is because vertical turbulent mixing becomes limited with increased stability, and non-local phenomena, like gravity waves and inertial oscillations, may occur.* "

**L74-75: I do not understand how this ties in with the main objective. Is it possible to reword the objective stated at the beginning of this paragraph to include climate models?**
*We have added a sentence so it is clear from the beginning of the paragraph how the eddy-covariance intercomparison and the comparison with the bulk estimates and the regional climate models link together.*

*"The aim of this study is to address this knowledge gap, and further, to determine the accuracy and validate EC measurements at a high elevation on a polar ice sheet. Using the validated EC measurements, we aim to improve the local bulk flux estimates and evaluate estimated fluxes from RCMs."*

**Figure 1: If possible, increase the resolution of this figure. A view of the footprint of the EC sensors would be useful. Also, there seems to be a fine-wire thermocouple on the IRGASON, but not on the other devices. Is it then the sonic temperature that is used to calculate the sensible heat flux? Please add these clarifications to the text.**

*The resolution of the figure has been increased. A view of the footprint and the landscape in the windward direction is provided below. The footprint area during times of regular wind direction included only an undisturbed snow surface (clean snow area). The reviewer is correct that there is a fine-wire thermocouple on the IRGASON, but this was not used in this*

*study. So the temperature used for the sensible heat flux is indeed the sonic temperature, which has now been clarified in the text.*

*"For all three EC systems, the air temperature measured by the sonic anemometer is used."*

[Figure]

**L87: Define 'clean-snow' area.**
*The clean-snow area refers to an area besides the ice-core drilling campsite where people are instructed not to walk. Therefore, ensuring that the snow in this area remains undisturbed. This has now been clarified in the text.*

*"The clean-snow area is a designated limited-access area oriented away from camp in the direction of the main wind direction to ensure undisturbed snow conditions."*

**L89-90: Why was the period from 28 May to 31 July 2019 used? Please explain.**
*The reason for this period is due to the availability of the necessary infrastructure at the site during the summer period, since the camp is only operated during the summer season.*

**L93: At this stage of the paper, it is not clear why the period from 2016 to 2019 is mentioned for the model comparison.**
*The aim of the model comparison is to show a comparison between observations and regional climate models on a longer timescale of several years. This specific period is chosen based on the availability of the different datasets since the PROMICE weather station was installed in 2016, and the MAR simulation runs until the end of 2019. This explanation has been added to the manuscript.*

*"A model comparison is done for the 4-year period from 2016-2019, during which both data from the PROMICE AWS and MAR simulation are available."*

**L95: Is it possible to better describe the observed wind regime, beyond the typical values of wind speeds and directions? Are there any katabatic winds?**
*As mentioned in the general comments, we have added more information about the katabatic winds to this section.*

*"EastGRIP is located in the north-eastern part of the Greenland Ice Sheet (Fig. 1a), in the katabatic wind zone, and approximately 350 km NNE of Summit, the highest point of the ice sheet."*

*"The mean windspeed was 4.5 m s$^{-1}$, with a maximum of 13.7 m s$^{-1}$, an average direction of 254°, and strong directionality due to the katabatic winds (Fig. 1b)."*

*"The average wind direction during this 4-year period was 242°, with 74 % of the time the wind direction falling within the 200 − 280° sector."*

**Section 2.2 and following: For the whole document, always present the three devices in the same order, to make it easier to follow.**
*The reason for presenting the devices in different orders is that for the introduction of the instruments, the order in which the systems are set up is used. But for the results, the EC-Irgason is used as reference system. To make this clearer, we have now explicitly added that the EC-Irgason is the reference system in the manuscript.*

*"The second EC system is the IRGASON (Campbell Scientific), hereafter used as reference system, which is a combined sonic anemometer and open-path gas analyser (Fig. 1e)"*

**L100: How and how often were these devices calibrated? Same question for the radiometer used to calculate the (very crucial) surface temperature.**
*For calibration of the PROMICE sensors, see Fausto et al. (2021). All three EC systems were calibrated in the factory. For the KH20, it is known that although the absolute humidity drifts over time, it still produces accurate flux measurements (Campbell Scientific, 2021). Humidity measurements from the IRGASON have been compared against a factory-calibrated PICARRO (Wahl et al, 2021) and the PROMICE humidity measurements. The Li-7500 was calibrated in the lab in December 2017, following LI-COR's recommended practice for a zero and span calibration for both the $H_2O$ and $CO_2$ measurements (Li-COR, 2004).*

**L111-113: This should have been mentioned earlier.**
*The longer availability of the EC-KH20 is now mentioned directly after the introduction of the instrument.*

*"The first EC system is a combination of a CSAT3 sonic anemometer (Campbell Scientific) and a Krypton Hygrometer 20 (KH20, Fig. 1d, Campbell Scientific). Besides 2019, this EC system was also deployed at EastGRIP during the summers of 2016, 2017 and 2018 (Steen-Larsen et al., a, b, 2022)."*

**L125: Mention that this is saturation with respect to ice. What is the validity of this hypothesis?**
*Following the approach of Fausto et al. (2021), we assume that the saturation is with respect to ice since we are on top of an ice sheet with very few days of temperatures above 0°C. While we agree that theoretically supercooled liquid could exist down to -20°C, we note the presence of ice crystals in the air, which would serve as heterogeneous nucleation nuclei.*

*"The surface specific humidity is determined using the surface temperature and assuming saturated conditions relative to ice."*

**L147: Presentation of Andreas' (1987) formulation would be useful.**

*The presentation of the Andreas parameterisation has been added as an appendix to the manuscript.*

**L150-151: I do not understand the changes resulting from the 'physics cycle CY47R.1' update. Is it possible to explain the highlights?**

*The physics cycle CY47R.1 refers to the physical parameterisations of the Integrated Forecast System (IFS) cycle 47r1. The complete list of upgrades coming with the updated physics cycle is described in Van Dalum et al., 2024. A summary of the upgrades coming with this physics cycle has been added to the manuscript:*

*"The upgraded physics cycle constitutes changes in the precipitation, convection, turbulence, aerosol and surface energy exchange schemes. RACMO 2.4 now uses the IFS radiation physics module ecRad, the new cloud scheme has more prognostic variables, and a multilayer snow module for non-glaciated regions is introduced. A fractional land–ice mask, as well as new and updated climatological data sets (such as aerosol concentrations), are used."*

**Equation 1a: I suggest removing the minus sign and writing Ts - T.**

*The suggested edit has been implemented*

**Equation 1b: I suggest removing the minus sign and writing qs - q.**

*The suggested edit has been implemented*

**Equation 4b: why not use the specific humidity q instead of a?**

*The covariance of the vertical windspeed and humidity provided by the processing software TK3 uses the absolute humidity instead of the specific humidity. Several correction steps (e.g. despiking and planar fit) are applied to the raw EC data before providing the covariance. The absolute and specific humidity are related via the air density, which is non-constant over time; therefore, a covariance of the specific humidity would need to be recalculated with the specific humidity. Since the TK3 software does not provide this option, the recalculation, including the corrections of the raw data, would need to be done manually. For simplicity and consistency in data processing with 4a and 4c, it was therefore chosen to convert the averaged specific humidity to absolute humidity instead.*

**L275-280: again - what was the calibration strategy (zero and span) for this instrument?**

*As also mentioned in the previous question, the Li-7500 was calibrated in the lab in December 2017, following LI-COR's recommended practice for a zero and span calibration for both the $H_2O$ and $CO_2$ measurements (Li-COR, 2004).*

**Figure 2 and equivalent: add a white box under the performance metric values and add the units on the RMSE.**

*We have implemented the suggested edits.*

**L298: How have the values of z0m, z0q and z0t been optimized?**

*Two approaches were used for this. The first is using the Andreas parameterisation to obtain $z_{0t}$ and assuming $z_{0q}=z_{0t}$. The second is using the Andreas parameterisation for both $z_{0t}$ and $z_{0q}$. For a discrete number of $z_{0u}$ (1e-9, 1e-8, 1e-7, 1e-6, 5e-6, 1e-5, 5e-5, 1e-4, 1e-3), the*

*slopes of the correlation between the computed bulk and measured EC flux (similar to figure 5) were computed where a slope close to 1 indicates a correctly simulated diurnal flux amplitude. In the figure below the slopes of the correlation between the bulk method and the EC are shown for the range of $z_{0u}$ values. The figure shows that there is no optimised $z_{0u}$ value using the Andreas parameterisation for both approaches that is suitable for both the LHF and the SHF. That is why we ended up optimising the three roughness lengths separately.*

[Figure]

**L299: 5.7e-7**
*We correct the missing exponential.*

**References:**

Andreas, E. L.: A theory for the scalar roughness and the scalar transfer coefficients over snow and sea ice, Boundary-Layer Meteorology, 38, 159–184, https://doi.org/10.1007/BF00121562, 1987.

Campbell Scientific: KH20 Krypton Hygrometer, URL https://s.campbellsci.com/documents/us/manuals/kh20.pdf, 2021.

Fausto, R. S., Van As, D., Mankoff, K. D., Vandecrux, B., Citterio, M., Ahlstrøm, A. P., Andersen, S. B., Colgan, W., Karlsson, N. B., Kjeldsen, K. K., Korsgaard, N. J., Larsen, S. H., Nielsen, S., Pedersen, A. Ø., Shields, C. L., Solgaard, A. M., and Box, J. E.: Programme for Monitoring of the Greenland Ice Sheet (PROMICE) automatic weather station data, Earth System Science Data, 13, 3819–3845, https://doi.org/10.5194/essd-13-3819-2021, 2021.

Li-COR: LI-7500 Open Path CO2/H2O Analyzer — Instruction Manual, https://licor.app.boxenterprise.net/s/ij79q7adnx7ozr1r1yil, revision 4, 2004.

Pickell, D. J., Hawley, R. L., and LeWinter, A.: Spatiotemporal patterns of accumulation and surface roughness in interior Greenland with a GNSS-IR network, The Cryosphere, 19, 1013–1029, https://doi.org/10.5194/tc-19-1013-2025, 2025.

Scanlan, K. M., Rutishauser, A., and Simonsen, S. B.: Greenland Ice Sheet surface roughness from Ku- and Ka-band radar altimetry surface echo strengths, The Cryosphere, 19, 1221–1239, https://doi.org/10.5194/tc-19-1221-2025, 2025.

Van Dalum, C. T., Van de Berg, W. J., Gadde, S. N., Van Tiggelen, M., Van der Drift, T., Van Meijgaard, E., Van Ulft, L. H., and Van den Broeke, M. R.: First results of the polar regional climate model RACMO2.4, EGUsphere, 2024, 1–36, https://doi.org/10.5194/egusphere-2024-895, 2024.

Van Tiggelen, M., Smeets, P. C. J. P., Reijmer, C. H., Wouters, B., Steiner, J. F., Nieuwstraten, E. J., Immerzeel, W. W., and van den Broeke, M. R.: Mapping the aerodynamic roughness of the Greenland Ice Sheet surface using ICESat-2: evaluation over the K-transect, The Cryosphere, 15, 2601–2621, https://doi.org/10.5194/tc-15-2601-2021, 2021.

Wahl, S., Steen-Larsen, H. C., Reuder, J., & Hörhold, M. (2021). Quantifying the stable water isotopologue exchange between snow surface and lower atmosphere by direct flux measurements. Journal of Geophysical Research: Atmospheres, 126, e2020JD034400. https://doi.org/10.1029/2020JD034400

---

## Author Comment (AC2)

Reply to RC2: 'Comment on egusphere-2025-711', Anonymous Referee #2, 25 Apr 2025. In the following text, the Referee's comments are reported in bold text and the author's answers are noted in italics. The edited text in the manuscript is in green.

We thank the reviewer for the positive and constructive feedback, and we agree with the summary provided by the reviewer. We believe we have addressed all the comments and will go through them step by step below.

**This manuscript compares the turbulent fluxes estimated by 3 Eddy-Covariance instruments over one summer at the EastGRIP site in Greenland, with a view of estimating the quality and uncertainties of such estimations in polar context. These data are also used in conjonction with ancillary data from AWS to derive year-round estimates of turbulent fluxes and compare them to outputs from high-resolution regional climate models.**

**This contribution is of high interest as, as underlined by the authors, estimations of turbulent fluxes are rare in polar environments, while modelling uncertainties are high. The manuscript is very neatly written and illustrated.**

**Still I think that some important points need to be addressed before publication :**

**Main comments :**

**My main concern is directed to the hypothesis of similar roughness lengths for winter and summer (e.g. L308-310). This hypothesis is not justified in the manuscript. Actually, it is questionned by the authors themselves. The discussion around this hypothesis, evidences in favor of or agaist it, and/or ways to circumvent it, is very short in the manuscript. It needs to be enhanced as an important part of the results relies on it (esp. the comparison to the RCM and the assessment of the sign of the sublimation/deposition contribution to the surface mass balance in Greenland). Also, the sensitivity of the results to this very strong hypothesis needs to be shown.**

*We agree with the reviewer that the uncertainty from the roughness lengths can be better elaborated, and have conducted a sensitivity study to improve the uncertainty from the roughness lengths on the flux estimates in winter. For the sensitivity study, the bulk flux calculation has been redone using roughness lengths that are 1 and 2 orders of magnitude larger and smaller than the optimised $z_{0m}$, $z_{0t}$ and $z_{0q}$. Figure A1 indicates a larger spread in roughness lengths during summer, but this is likely the result of measurement uncertainty and the range of 1 and 2 orders of magnitude is chosen based on Van Tiggelen et al., 2023, Figure 3. This figure shows that the seasonal cycle of the roughness length for momentum at the edge of the Greenland Ice Sheet varies with an amplitude between 1 and 2 orders of magnitude. As the location is at the edge of the ice sheet, with crevasses and melt features, which is very different from EastGRIP, the two orders of magnitude should be a conservative estimate. The result of the sensitivity analysis is shown below and added to the supplementary material. The spread between the calculations with different roughness lengths during the summer and shoulder months clearly highlights the sensitivity of the flux calculation to the magnitude of the roughness length. However, little sensitivity to the roughness length can be seen during winter, shown by the lack of spread between the*

*different calculations. This confirms that the difference seen between the fluxes from the AWS and the RCMs in winter is mainly driven by the difference in near-surface atmospheric temperature gradient, and not the roughness length.*

*"However, as no EC measurements were conducted during the winter, we only evaluate the estimated bulk fluxes during the summer (daylight) period. We assume that using the improved roughness lengths the calculated bulk fluxes provide reliable estimates during the winter as well. This is supported by a sensitivity study using roughness lengths up to one and two orders of magnitude smaller or larger than the original values (following Van Tiggelen et al., 2023), showing that the exact value of the roughness length has limited influence on the estimated flux during the winter (see Supplementary Material S5)."*

[Figure]

*Figure S6. Seasonal cycle of the (a) LHF and (b) SHF from figures 6a and 7a, respectively, with the shaded areas indicating the LHF and SHF calculated from the PROMICE AWS data, using values for $z_{0,u}$, $z_{0,q}$ and $z_{0,t}$ that are one (medium blue) and two (light blue) orders of magnitude (OoM) smaller or larger, respectively, compared to the EC-derived roughness length values (see table S3).*

**Secondly, it seems that blowing snow sublimation can greatly modify the magnitude of the total sublimation/deposition at the annual scale, and even change the bulk contribution to the mass balance from net gain (deposition) to net loss. The authors note that the estimation of turbulent fluxes with the bulk method used in winter, is not valid during blowing snow events (L326). Furthermore, the models diverge in their accounting (or not) of this process. I would strongly suggest to exclude the periods with blowing snow events from the direct model-to-data comparisons in section 4.3 for a fairer assessment of model performances. If no better, RACMO2.3 outputs could possibly be used for a first diagnostic of the main blowing snow events.**

*We agree with the reviewer that the accounting of the blowing snow and the corresponding blowing snow sublimation is important. However, filtering out blowing snow events from the datasets is a non-trivial task. In RACMO, blowing snow (which includes both blowing and drifting snow, Gadde and Van den Berg, 2024) occurs when the friction velocity is higher than the threshold friction velocity. As EastGRIP is relatively windy, this means that in the model some form of blowing snow occurs over half of the time, especially in winter, and filtering this out would remove approximately 85% of the winter time data (see table below). Due to uncertainties in the blowing snow scheme, and therefore also blowing snow sublimation, filtering out the strongest values of blowing snow sublimation in the model does*

*not guarantee that this is also the case for the weather station data. As blowing snow events cannot be directly identified from the AWS data, filtering based on the wind speed is possible, but again does not guarantee that all blowing snow is removed from the model data (see also the figure below). Therefore, we think that a simple filtering approach might introduce more uncertainties than would be resolved, and a more thorough approach would be needed, which falls outside the scope of this paper.*

[Figure]

*Scatter plot of the blowing snow sublimation in RACMO2.4 (daily values) against the windspeed from the PROMICE AWS (daily sum of hourly values) from 2016 to 2019.*

*Average percentage of days per month during the model comparison period in RACMO2.4 where no blowing snow occurs.*

| Month | 1 | 2 | 3 | 4 | 5 | 6 | 7 | 8 | 9 | 10 | 11 | 12 |
|---|---|---|---|---|---|---|---|---|---|---|---|---|
| % days without blowing snow | 15 | 11 | 25 | 17 | 50 | 62 | 69 | 60 | 27 | 16 | 13 | 15 |

**Finally, I think the discussion regarding the model and observation comparison, could benefit from an enhanced context regarding the known model biases over polar and/or snow-covered environments. L 432 incriminates « the atmospheric processes driving the surface gradients» in the models, but surface processes are also likely at stake, esp. in the presence of snow. Fettweis et al. 2017 highlight some of them for MAR over Greenland, and they should be mentioned. With respect to that, Lapo et al., 2015 noted the important role of stability corrections in amplifying model cold biases over snow surfaces, esp. in conjuction with a negative bias in incoming LW radiations (which seems to be affecting e.g. the MAR model ; Fettweis et al., 2017). Could this play a role here too ? This possible source of model bias should be included in the discussion (Rudisill et al., 2024 could provide an intesting view on other bias sources, though more oriented towards mountain regions).**

*We thank the reviewer for this useful comment and have removed "atmospheric" from the processes, as it is indeed a combination of surface and atmospheric processes. As highlighted by the mentioned articles, there are a lot of processes involved in the near-surface temperature gradient, like the longwave radiation, stability functions, albedo, mixing and snow properties, which also feed back on each other. Investigating what exactly causes the systematically larger near-surface gradient in the models compared to the measurements would be a whole study on its own and therefore falls outside the scope of this paper. We have, however, expanded some of the known model biases, likely involved in too large near-surface temperature gradients. Known biases both for MAR and RACMO are the incoming longwave radiation, as also mentioned by Fettweis et al., 2017. Which, as mentioned in Lapo et al., 2015, can lead to a bias in the surface temperature. Lapo et al., 2015 also mention the influence of the stability corrections. Both MAR and RACMO use stability correction adapted for polar conditions. This likely falls in the general uncertainty of the bulk method, especially in the mentioned polar conditions. With regard to the stability, we hypothesise that katabatic mixing processes might be missed in the models, leading to overestimated stability.*

*"In fact, by correcting the MAR model with summer observations at EastGRIP, Dietrich et al. (2024) find the LHF to be a negative SMB contributor in their simulations. They propose that the difference in LHF between the model and observations during summer arises from a negative bias in downwelling longwave radiation, as also found by Fettweis et al. (2017), from the cloud scheme, while the winter bias may be caused by vertical mixing through katabatic winds that is not represented in the model (Dietrich et al., 2024). Similar to Dietrich et al. (2024), we find that RACMO in winter also overestimates deposition compared to AWS observations at EastGRIP, probably a result of too low surface temperatures caused by a negative bias in incoming longwave radiation (Van Dalum et al., 2024) "*

**Minor comments :**

**L 8-10 : this is a lot of detailed numbers for an abstract, maybe choose 2 metrics out of the 4 presently described**

*We have adjusted the sentence and removed the correlation and RMSE.*

*"A comparison of the fluxes by the three systems demonstrates excellent agreement with an absolute bias of 0.2 W m⁻² and slopes between 1.01 and 1.16 for the LHF, and an absolute bias of less than 0.5 W m⁻² and slopes of 1.0 for the SHF"*

**L 87 : what is a clean-snow area ? Please define.**

*The clean-snow area refers to an area besides the ice-core drilling campsite where people are instructed not to walk. Therefore, ensuring that the snow in this area remains undisturbed. The following sentence is added to the section:*

*"The clean-snow area is a designated limited-access area oriented away from camp in the direction of the main wind direction to ensure undisturbed snow conditions."*

**L 91 : the maximum on wind speed can highly differ depending on time-averaging procedure. Is this a maximum of 10-min data, hourly data, or instantaneous maximum ?**

*The maximum windspeed is the maximum of the hourly averaged data. This clarification has been added to the section.*

*"The average wind direction during this 4-year period was 242°, with 74 % of the time the wind direction falling within the 200 − 280° sector. All values are based on hourly averaged data."*

**L 151 : the major changes attached to the upgrade to CY47R.1 should be described in a nutshell**

*The complete list of upgrades coming with the updated physics cycle is described in Van Dalum et al., 2024. A summary of the upgrades coming with this physics cycle has been added to the manuscript:*
*"The upgraded physics cycle constitutes changes in the precipitation, convection, turbulence, aerosol and surface energy exchange schemes. RACMO 2.4 now uses the IFS radiation physics module ecRad, the new cloud scheme has more prognostic variables, and a multilayer snow module for non-glaciated regions is introduced. A fractional land–ice mask, as well as new and updated climatological data sets (such as aerosol concentrations), are used."*

**L202-203 : site-specific roughness lengths would likely depend on snow conditions and hence vary at least sesonally ? (see 1st main comment)**

*We agree with the reviewer that there is likely a seasonal cycle in the roughness length as a result of changing snow conditions (also shown in Van Tiggelen et al., 2023). However, as the instrumentation does not allow for obtaining roughness length estimates during winter (see also Review 1), a fixed roughness length has been used. To quantify the uncertainty of this assumption, we added a sensitivity study. See the response to the main concerns.*

**L289 : « Estimates of the LHF and SHF based on the bulk method in the PROMICE data product are overestimated (Fig. 5) ». Yet the biases mentioned on Fig5 are mostly negative, probably due to different sign convention. It would be clearer if it could be changed**

*We thank the reviewer for the sharp observation of the sign convention and have changed it, so it is now consistent.*

**Fig7a : the bulk estimate of sensible heat flux shows a bi-modal seasonal cycle, much different from all models, but this seems to me this is barely analysed. Can you comment on this ?**

*Looking into the variables driving the SHF (see figure below), it can be seen that the difference in near-surface atmospheric temperature gradient is what drives the increased SHF during the shoulder months. For this paper, the focus was on analysing the winter months, as this is the season with the most pronounced difference between the RCMs and*

the PROMICE data. Although we agree that this feature in the shoulder months is interesting, this fell outside the scope of the paper. We speculate that it might relate to the heat transfer coefficient and heat capacity of the snow and the interaction with the atmosphere during the transition to and from polar night to the sunlit period.

[Figure]

**L 365-366 : «However, systematic biases with the EC method, due to boundary layer characteristics in polar conditions cannot be ruled out. » This should be developped.**

*As mentioned in Section 3.1, the way the EC systems are set up should largely comply with the assumptions going into EC flux calculation and we have expanded on the characteristics of the polar conditions:*

*"However, systematic biases with the EC method, due to boundary layer characteristics in polar conditions cannot be ruled out, such as the influence of the katabatic wind maximum, and during (sub)meso motions (Lan et al., 2022), which both typically take place under very stable conditions and can cause the transport to deviate from a fully turbulence-dominated regime."*

**L397 : erroneous reference : Fig 8 a is probably meant ?**

*This is correct and we have checked and corrected the references in this section.*

**L405 : « Features in the synoptic-scale variability of the near-surface temperature gradient are observable in both AWS datasets. » It seems that these features are much more attenuated at the PROMICE station, esp for the second half of the period shown Fig 8, which may be an argument for frost on the LWup sensor ?**

*It is good to keep in mind that the temperature gradient of the PROMICE AWS is measured at approximately 2 m using a temperature sensor and at surface height using the longwave radiation. While the GC-Net AWS measured the temperature gradient over approximately 1.3 m using two temperature sensors. The attenuation could therefore potentially also be explained by the smaller distance over which the GC-Net measures. Where the smaller distance and the measurement uncertainty of the two sensors lead to a higher variability.*

**L 409 : words are likely missing in this sentence**

*We have changed the sentence, and it now reads:*

**L 415 : «  the air temperature is lower than the snow surface (e.g. 1st until the 10th of January 2019) » It seems to me that air temperature is actually mostly warmer than snow surface over this period…, which makes the discussion confusing. The whole section 5.3 should be checked carefully as the processes at stake are not straightforward and it is not easy to get the point of the authors. Maybe the section could be renamed « Limits of the PROMICE data during winter », and a contradictory time-period when a no-frost assumption can clearly be made, could be provided for comparison in the graphs ?**

*We thank the reviewer for this useful feedback and have changed the section title to "Caveats related to the PROMICE data in winter". The reviewer was correct that the dates were switched. This is now corrected and the section is rewritten so it is hopefully easier to follow:*

*"A second line of evidence supporting the radiation sensor measurements can be deduced from the shallow near-surface (10 cm) snow temperature, which is less prone to measurement uncertainties (Fig. 8c and 8d). Due to the thermal mass of the snow, it is not to be expected that the magnitude of the temperature gradient deduced from the near-surface snowpack temperature ($T_{2.6}$ − $T_{Snow}$, Fig. 8d) is equivalent to the temperature gradient deduced from the snow skin surface ($T_{2.6}$ − $T_{Surf}$, Fig. 8e). However, we note the general good agreement in the evolution of the sign of the temperature gradient based on the snowpack ($T_{2.6}$ − $T_{Snow}$) and skin surface temperature ($T_{2.6}$ − $T_{Surf}$). For example, when colder air is present, the air temperature is lower than the snow surface (e.g. 10th until the 15th of January 2019) and vice-versa (e.g. 1st until 10th of January). Interpretation is, however, not straightforward, as the thermal inertia of the snowpack leads to a delayed response to surface forcing, and snow temperature reflects a delayed and smoothened pattern of the air temperature, reflecting a mixture of heat exchange between air, surface and subsurface at longer time scales."*

*For this research, the choice was made to focus on comparing the observations from the PROMICE AWS with the RCMs during winter (January and December), since in this period the largest differences are seen. We have checked the PROMICE data from the other winter months, and the data from the other months is similar to the time period shown and described in Section 5.3. It is therefore not possible to do a direct case study comparison. We have, however, added the timeseries of the other winter months to the supplementary material and at the end of this review.*

*"Features in the synoptic-scale variability of the near-surface temperature gradient are observable in both AWS datasets (see Supplementary Material S7 for other winter months)."*

**Sect 5.4 / sect 5.3 : The whole analysis of the modelled vs observed near-surface temperature gradients in Sect 5.4 is based on an extract of the time-series that is precisely questionned regarding the observation of surface temperature one section before. Would it be possible to carry out this analysis over another period where surface temperature data would be less questionable ? Statistics of the occurrence of**

**such doubtfull PROMICE data would be valuable to assess the PROMICE winter data quality as a whole (Sect 5.3), and shed light on the results/interpretations. A feedback to the data exclusion mentioned in Sect 2.3 would be usefull for the reader's understanding.**

*As mentioned in the previous answer is the data from the other winter months similart, but we have repeated the analysis for the other winter months as well. In the figure below, a comparison of the near-surface atmospheric temperature gradient is shown of the PROMICE and GC-Net AWS and the three RCMs for all winter months (also added to the manuscript). Here, it can be seen that both the PROMICE and GC-Net AWS consistently measure a smaller temperature gradient during the winter months than the three RCMs.*

*"The independently observed temperature gradients by the single-level PROMICE and two-level GC-NET AWS are generally approximately similar and an order of magnitude smaller than the modelled temperature gradient (see Fig. 9i and Fig. 10 for all winter months)."*

*"With the tentative assumption that the observed surface temperatures are correct, the evaluated climate models simulate a too strong stability in this part of the ice sheet during winter. This is supported by a comparison of the near-surface temperature gradient measured by both the PROMICE and the two-level GC-Net AWS and the three RCMs over all winter months (Fig. 10)."*

*We have added a sentence to section 5.3 linking back to the data exclusion in section 3.2.*

*"The near-zero net longwave radiation sometimes coincides with low temperatures (T < −50 °C), conditions normally associated with a surface based temperature inversion rooted in longwave radiative cooling (Van den Broeke et al., 2004; Miller et al., 2017). Times when T < −50 °C are therefore also removed from the data (Sect 2.3)."*

[Figure]

*Figure 10. Boxplots of the near-surface atmospheric temperature gradients of the winter (DJ) months from the PROMICE, GC-Net AWS and three RCM's MAR, RACMO 2.3 and RACMO 2.4. The whiskers of the boxplots indicate the 5th-95th percentile, the box the 25th to 75th percentile, the thick line the median and the black dash (-) the mean. Note that the temperature gradients of the PROMICE AWS and the three RCMs are between 2 m and the surface, while the GC-Net is between two air temperature sensors spaced 1.3 m apart, with the lowest approximately 1 to 2 m above the surface.*

**References :**

Fettweis, X., Box, J. E., Agosta, C., Amory, C., Kittel, C., Lang, C., van As, D., Machguth, H., and Gallée, H.: Reconstructions of the 1900–2015 Greenland ice sheet surface mass balance using the regional climate MAR model, The Cryosphere, 11, 1015–1033, https://doi.org/10.5194/tc-11-1015-2017, 2017.

Gadde, S. and Van de Berg, W. J.: Contribution of blowing-snow sublimation to the surface mass balance of Antarctica, The Cryosphere, 18, 4933–4953, https://doi.org/10.5194/tc-18-4933-2024, 2024.

Lapo, K. E., L. M. Hinkelman, M. S. Raleigh, and J. D. Lundquist : Impact of errors in the downwelling irradiances on simulations of snow water equivalent, snow surface temperature, and the snow energy balance, Water Resour. Res., 51, 1649–1670, doi:10.1002/ 2014WR016259, 2015.

Rudisill, W., A. Rhoades, Z. Xu, and D. R. Feldman: Are Atmospheric Models Too Cold in the Mountains? The State of Science and Insights from the SAIL Field Campaign. *Bull. Amer. Meteor. Soc.*, 105, E1237–E1264, https://doi.org/10.1175/BAMS-D-23-0082.1, 2024.

Van Dalum, C. T., Van de Berg, W. J., Gadde, S. N., Van Tiggelen, M., Van der Drift, T., Van Meijgaard, E., Van Ulft, L. H., and Van den Broeke, M. R.: First results of the polar regional climate model RACMO2.4, EGUsphere, 2024, 1–36, https://doi.org/10.5194/egusphere-2024-895, 2024.

Van Tiggelen, M., Smeets, P. C. J. P., Reijmer, C. H., Van den Broeke, M. R., Van As, D., Box, J. E., and Fausto, R. S.: Observed and Parameterized Roughness Lengths for Momentum and Heat Over Rough Ice Surfaces, Journal of Geophysical Research: Atmospheres,675 128, e2022JD036 970, https://doi.org/https://doi.org/10.1029/2022JD036970, e2022JD036970 2022JD036970, 2023.

Time series other winter months:

[Figure]

*Figure S7. Time series of winter AWS observations in December 2016 with (a) the net longwave radiation, (b) the calculated sensible and latent heat flux, using the $z_{0,m}$, $z_{0,q}$ and $z_{0,t}$ values derived from the EC-Irgason, (c) the air temperature at 2.0 m above the surface and the snow temperature approximately 10 cm below the surface, (d) the temperature difference between the 2.0 m and snow temperature and (e) the PROMICE temperature difference between 2.0 m and the surface, determined via the longwave radiation and (f) the GC-Net temperature difference between 3.3 m and 2.0 m. Note that the complete dataset is shown for this figure, including the data below −50°C.*

[Figure]

*Figure S8. Time series of winter AWS observations in January 2017 with (a) the net longwave radiation, (b) the calculated sensible and latent heat flux, using the $z_{0,m}$, $z_{0,q}$ and $z_{0,t}$ values derived from the EC-Irgason, (c) the air temperature at 2.0 m above the surface and the snow temperature approximately 10 cm below the surface, (d) the temperature difference between the 2.0 m and snow temperature and (e) the PROMICE temperature difference between 2.0 m and the surface, determined via the longwave radiation and (f) the GC-Net temperature difference between 3.2 m and 2.0 m. Note that the complete dataset is shown for this figure, including the data below −50°C.*

[Figure]

*Figure S9. Time series of winter AWS observations in December 2017 with (a) the net longwave radiation, (b) the calculated sensible and latent heat flux, using the $z_{0,m}$, $z_{0,q}$ and $z_{0,t}$ values derived from the EC-Irgason, (c) the air temperature at 2.2 m above the surface and the snow temperature approximately 10 cm below the surface, (d) the temperature difference between the 2.2 m and snow temperature and (e) the PROMICE temperature difference between 2.2 m and the surface, determined via the longwave radiation and (f) the GC-Net temperature difference between 2.9 m and 1.6 m. Note that the complete dataset is shown for this figure, including the data below −50°C.*

[Figure]

*Figure S10. Time series of winter AWS observations in January 2018 with (a) the net longwave radiation, (b) the calculated sensible and latent heat flux, using the $z_{0,m}$, $z_{0,q}$ and $z_{0,t}$ values derived from the EC-Irgason, (c) the air temperature at 2.2 m above the surface and the snow temperature approximately 10 cm below the surface, (d) the temperature difference between the 2.2 m and snow temperature and (e) the PROMICE temperature difference between 2.2 m and the surface, determined via the longwave radiation and (f) the GC-Net temperature difference between 2.9 m and 1.6 m. Note that the complete dataset is shown for this figure, including the data below −50°C.*

[Figure]

Figure S11. Time series of winter AWS observations in December 2018 with (a) the net longwave radiation, (b) the calculated sensible and latent heat flux, using the $z_{0,m}$, $z_{0,q}$ and $z_{0,t}$ values derived from the EC-Irgason, (c) the air temperature at 2.4 m above the surface and the snow temperature approximately 10 cm below the surface, (d) the temperature difference between the 2.4 m and snow temperature and (e) the PROMICE temperature difference between 2.4 m and the surface, determined via the longwave radiation and (f) the GC-Net temperature difference between 2.4 m and 1.1 m. Note that the complete dataset is shown for this figure, including the data below −50°C.

[Figure]

*Figure S12. Time series of winter AWS observations in December 2019 with (a) the net longwave radiation, (b) the calculated sensible and latent heat flux, using the $z_{0,m}$, $z_{0,q}$ and $z_{0,t}$ values derived from the EC-Irgason, (c) the air temperature at 2.3 m above the surface and the snow temperature approximately 10 cm below the surface, (d) the temperature difference between the 2.3 m and snow temperature and (e) the PROMICE temperature difference between 2.3 m and the surface, determined via the longwave radiation and (f) the GC-Net temperature difference between 2.2 m and 0.9 m. Note that the complete dataset is shown for this figure, including the data below −50°C.*